# Impact of disease-associated chromatin accessibility QTLs across immune cell types and contexts

## Graphical abstract

## Authors

Zepeng Mu (牟泽鹏), Haley E. Randolph, Raúl Aguirre-Gamboa, ..., Daniel E. Kaufmann, Luis B. Barreiro, Yang I. Li

## Correspondence

lbarreiro@uchicago.edu (L.B.B.), yangili1@uchicago.edu (Y.I.L.)

## In brief

Mu et al. performed single-cell chromatin accessibility quantitative trait locus (caQTL) mapping in human peripheral blood mononuclear cells and found that caQTL explain roughly 50% more disease genome-wide association study loci compared to eQTLs. They also showed evidence that widespread sharing of caQTLs among immune cells obscures causal cellular context identification, highlighting the need to study regulatory QTLs in disease-relevant contexts.

## Highlights

- Unified scATAC map of healthy and COVID-19 PBMC in 48 donors with ∼280,000 cells

- Topic modeling reveals CD8 TEM chromatin state continuum associated with COVID-19

- Single-cell caQTLs explain 50% more GWAS loci than eQTLs

- caQTLs sharing across cellular contexts obscure causal context identification

 Mu et al., 2026, Cell Genomics 6, 101061
January 14, 2026 © 2025 The Authors. Published by Elsevier Inc.

## Article

# Impact of disease-associated chromatin accessibility QTLs across immune cell types and contexts

Zepeng Mu (牟泽鹏),[1,2] Haley E. Randolph,[1,3] Raúl Aguirre-Gamboa,[4] Ellen Ketter,[5] Anne Dumaine,[4] Veronica Locher,[6] Cary Brandolino,[4] Xuanyao Liu,[1,4,7] Daniel E. Kaufmann,[8,9] Luis B. Barreiro,[1,4,6,7,10,*] and Yang I. Li[1,4,7,10,11,*]

[1]Committee on Genetics, Genomics & Systems Biology, University of Chicago, Chicago, IL, USA
[2]Center for Data Sciences, Brigham and Women's Hospital, Harvard Medical School, Boston, MA, USA
[3]Department of Pediatrics, Columbia University Irving Medical Center, New York, NY, USA
[4]Section of Genetic Medicine, Department of Medicine, University of Chicago, Chicago, IL, USA
[5]Committee on Microbiology, University of Chicago, Chicago, IL, USA
[6]Committee on Immunology, University of Chicago, Chicago, IL, USA
[7]Department of Human Genetics, Department of Medicine, University of Chicago, Chicago, IL, USA
[8]Division of Infectious Diseases, Department of Medicine, University Hospital and University of Lausanne, Lausanne, Switzerland
[9]Centre de Recherche du CHUM (CRCHUM) and Département de Médecine, Université de Montréal, Montreal, QC, Canada
[10]CZ Biohub Chicago, Chicago, IL, USA
[11]Lead contact
*Correspondence: lbarreiro@uchicago.edu (L.B.B.), yangili1@uchicago.edu (Y.I.L.)

## SUMMARY

Only one-third of immune-associated genome-wide association study (GWAS) loci colocalize with expression quantitative trait loci (eQTLs), leaving most mechanisms unresolved. To address this, we created a unified single-cell chromatin accessibility (scATAC) map of ~280,000 peripheral immune cells from 48 individuals, including 20 COVID-19 patients. Topic modeling of scATAC data identified continuous cell states and revealed disease-relevant cellular contexts. We identified 37,390 chromatin accessibility QTLs (caQTLs) at 10% false discovery rate and observed extensive sharing of caQTLs, with <20% confined to a single context. Notably, caQTLs explained ~50% more GWAS loci compared to eQTLs, nominating putative causal genes for some unexplained loci. Yet most GWAS-colocalizing caQTLs lacked eQTL support, limiting causal inference from chromatin data alone. Thus, while caQTLs can improve GWAS interpretation, robust mechanistic insights require integration with gene expression and other functional evidence. Our work underscores that cellular context is critical for regulatory variant interpretation and emphasizes the need to map genetic effects in disease-relevant cell states.

## INTRODUCTION

A major goal in complex trait genomics is to understand the biological mechanisms of trait-associated variants. To this end, a general approach has been to map molecular quantitative trait loci (molQTLs) in one or more cell types or states and then colocalize these molQTLs with genome-wide association study (GWAS) loci. GWAS loci colocalized with a molQTL are then often considered to be "explained." To date, molQTLs of gene-expression levels (expression QTLs [eQTLs]) have been the focus of nearly all studies. Although eQTLs have greatly improved our ability to identify genes and contexts that are impacted at many GWAS loci, over half of GWAS signals remain unexplained for most complex traits.[1]

Several groups have proposed that standard eQTL analyses from bulk samples generally identify large but unimportant genetic effects that are shared across many cell types.[2] Consistent with this view, an analysis of GTEx eQTLs—the largest collection of eQTLs, which covers over 55 human organs—revealed that *cis*-eQTLs only mediate ~11% of trait heritability on average.[3] These findings are often interpreted to suggest that many GWAS variants function through cell-type- or context-specific effects on gene regulation, motivating searches for eQTLs in specific cell types and contexts (i.e., cell subtypes/state, disease conditions) that may be more relevant to the traits of interest. Indeed, several studies have now identified cell-type-specific and highly transient genetic effects on gene-expression level that colocalize with association signals at GWAS loci.[4,5] Even so, each study only contributes to a tiny number of additional co-localizations, raising questions as to whether this approach is effective.

More recently, two studies mapped chromatin phenotype QTLs (cQTLs) and found that cQTLs substantially increased the fraction of GWAS loci that colocalizes with a molQTL.[6,7] It

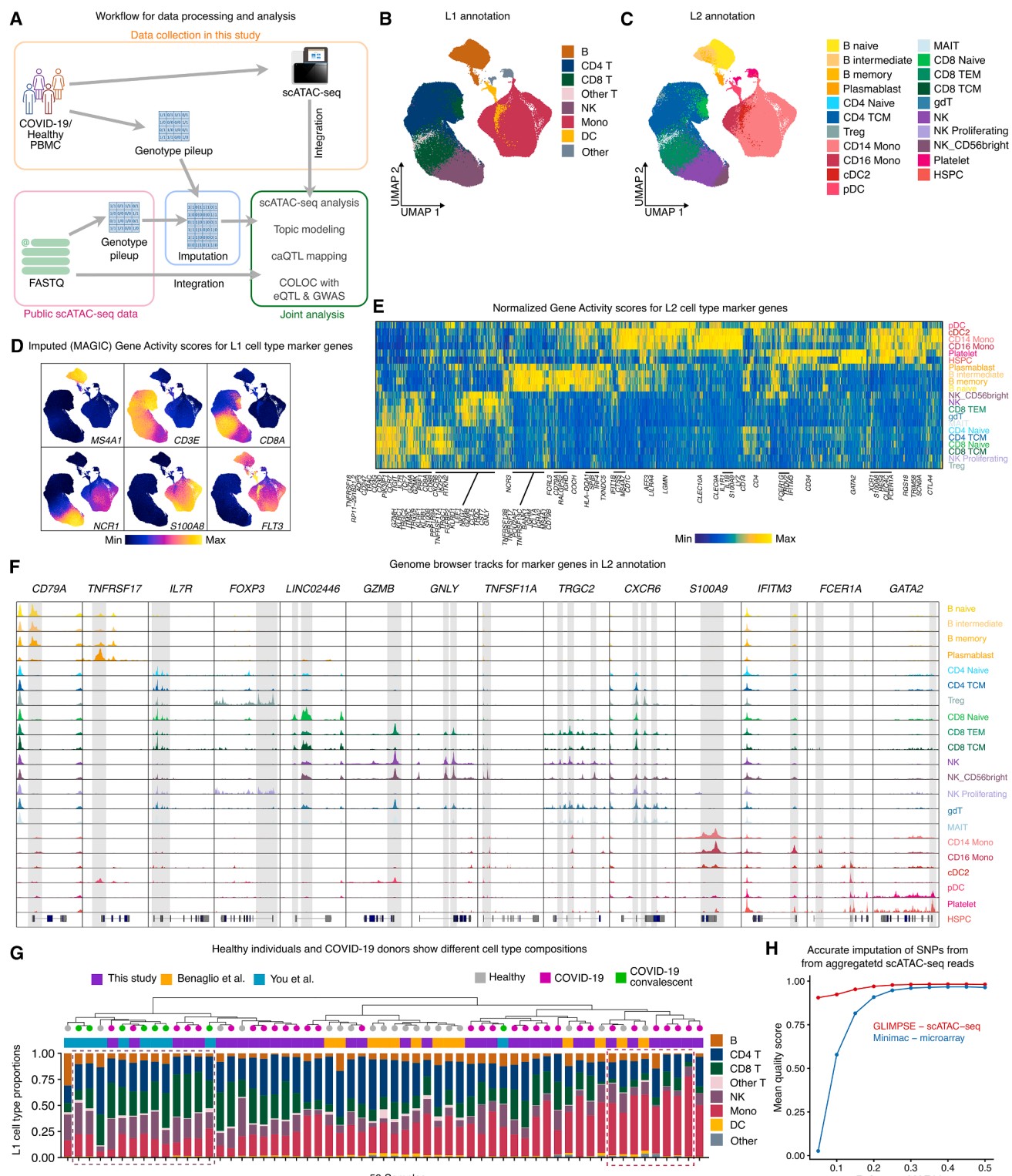

**Figure 1. An integrated map of scATAC-seq of PBMC from 20 donors with active COVID-19, 19 COVID-19 convalescent donors, and 20 healthy controls**

(A) Schematic representation of sample collection, data integration, and analysis workflow. Some icons are from bioicons.com.

(B) UMAP embedding of all cells from three integrated studies, colored by L1 annotation of seven common immune-cell types.

(C) The same UMAP embedding as in (B), colored by 21 immune-cell types and subtypes in L2 annotation.

*(legend continued on next page)*

is unclear why the rates of cQTL colocalization are larger than that of eQTLs, especially given that cQTLs are expected to impact gene-expression levels in their causal path to influence human traits. Still, these findings, along with earlier reports that trait heritability explained by eQTL SNPs (11%–14%)[8] are generally smaller than that explained by SNPs in enhancer and promoter regions (24%–79%),[9] suggest that mapping QTLs for chromatin-level phenotypes, such as chromatin accessibility QTLs (caQTLs), may help unravel the genetic mechanism of unexplained GWAS loci.

To test this strategy, we constructed a unified single-cell chromatin accessibility (scATAC) atlas of peripheral blood mononuclear cells (PBMCs) from 48 individuals across three independent studies.[10,11] Although our study is not focused on COVID-19, our data include 20 COVID-19 patients and eight convalescent donors. After integrating these datasets, we developed novel computational approaches to systematically assess how immune-related GWAS variants influence chromatin accessibility across cell types, states, and dynamic trajectories. When we compared the caQTLs we mapped with cell-type-resolved eQTLs, we found that a substantial fraction of chromatin accessibility changes had no corresponding gene-expression effects in the same cell types or contexts. This discrepancy was not due to limited statistical power but instead correlated with the absence of enhancer-promoter interactions, suggesting that many genetic effects on chromatin accessibility may not be functionally consequential in most cell types.

Notably, many immune GWAS loci that colocalized with caQTLs failed to colocalize with any eQTL in any assayed cell types, suggesting that critical disease-relevant cellular contexts remain uncharacterized in existing eQTL datasets. Our findings underscore the need to expand eQTL mapping into more disease-relevant cell states to decipher the regulatory basis of complex genetic associations.

## RESULTS

### A harmonized map of chromatin accessibility in immune cells from 59 samples

To build a detailed chromatin accessibility atlas of PBMCs, we integrated scATAC sequencing (scATAC-seq) data from 59 samples across three studies,[10,11] including 20 active COVID-19 patients, 19 convalescent donors, and 20 healthy controls from diverse genetic ancestries (Figure 1A and Table S1). After quality filtering (Figures S1A and S1B) and batch correction (STAR Methods), we retained 282,424 high-quality cells. We annotated cell types using a reference single-cell RNA sequencing[12] dataset (Figure S1C), resulting in two levels of granularity: seven major types (L1) for QTL mapping and 21 subtypes (L2) for

fine-grained analysis (Figures 1B and 1C). We defined a unified set of 327,746 chromatin peaks and confirmed annotation quality by visualizing high gene-activity (GA) scores for canonical cell-type markers (e.g., *MS4A1* in B cells and *CD3E* in T cells) (Figures 1D–1F). While overall cell-type compositions were similar across donor groups, we observed expansions of natural killer (NK) cell or monocyte populations in two COVID-19 patient subgroups (Figure 1G), consistent with previous reports.[11,13] At the L2 resolution, we detected elevated proportions of memory B cells in COVID-19 patients (Figure S1D). While L2 annotation provides enhanced cellular-state resolution, its utility for quantitative analysis is limited because of reduced statistical power due to smaller cell counts per state. Consequently, we employed L1 annotations for our primary QTL mapping to maintain statistical power, reserving L2 annotations for specific analyses where finer cellular resolution was required.

Finally, we imputed high-quality genotypes for all individuals by aggregating scATAC-seq reads, achieving high concordance with microarray-based genotypes (Figures 1H and S1E–S1I) and enabling integrated QTL mapping (STAR Methods).

Altogether, we constructed a map of accessible chromatin from 282,424 PBMCs from 48 individuals (59 samples) with high-quality harmonized genotype information for all individuals, enabling fully integrated downstream analysis. Imputation using aggregated scATAC-seq reads offers high-quality genotype information, and our workflow (data and code availability) can be easily adopted for future population-scale scATAC studies.

### Topic analysis of chromatin accessibility defines cell type and state programs

Single-cell genomics data can capture gene expression in rare cell types as well as transitional cell states. However, typical single-cell data analysis aggregates cells into discrete clusters, masking heterogeneity among cells within the same cluster. As an alternative to clustering, we applied topic modeling to our scATAC-seq data. Topic modeling represents each cell as a grade of membership (referred to as "loadings" hereafter) to inferred topics.[14,15] Each topic captures an axis of variation in the data, which may represent cell types, contexts, or biological processes. This allows us to identify peaks with differential accessibility across topics and to measure the importance of a peak to each topic (referred to as "scores" hereafter).[16] As such, peaks with the highest scores in each topic often reveal their associated biological functions.

To reach a deeper understanding of cell-state variations in our scATAC data and complement our cluster-based analysis, we applied fastTopics[17] to our scATAC count matrix and built models for 6–20 total topics (referred to as "k" hereafter).

(D) Gene-activity scores of marker genes in the seven common immune-cell types. Scores were imputed with MAGIC for visualization purposes.

(E) Heatmap for marker genes for cell subtypes in L2 annotation.

(F) Color-coded genome browser tracks of aggregated scATAC-seq reads in genomic loci around marker genes. Shaded regions highlight cell-type-specific chromatin accessibility regions.

(G) Estimates of cell-type compositions in L1 annotation for all samples. Samples are clustered by distances in scaled cell-type compositions. Samples with expanded NK or monocyte populations are highlighted in dashed boxes.

(H) Comparison of imputation quality (INFO) score from DNA microarray using Minimac4 and aggregated scATAC-seq reads using GLIMPSE stratified by reference MAF bins in the study by Benaglio et al.[10]

Ultimately, we selected a 20-topic-model (k = 20), as it captured major cell types and recognizable cell states (Figures 2A and S2). Several topics represented technical variation (e.g., k2 correlated with transcription start site [TSS] enrichment, Figure 2B) and were excluded from biological interpretation (STAR Methods; Figures S3A, S3B, and S4).

To functionally annotate the different topics, we derived gene-level scores in each topic from the peak-level scores (Figure S5). This allowed us to identify a set of genes driving each topic, and we observed well-known cell-type markers among the highest-scoring genes: *EBF1* and *CD83* for naive B cells (k1); *CD27* and *TNFSF9* for memory B cells (k11); *CD247* and *ZBTB16* for NK cells (k8); *S1PR5*, *KLRD1*, *PRF1*, and *TBX21* for cytotoxic CD8 T cells (k3); and *ICOS* and *CTLA4* for regulatory T (Treg) cells (k20) (Figure 2C and Table S2). The biological relevance of each topic was further validated by significant enrichment of relevant Gene Ontology (GO) terms among high-scoring genes (Figure S3C and Table S3) and of known immune-cell-type-specific transcription factor (TF) binding motifs in their top chromatin peaks (Figure S3D and Table S4).

Cell-state transitions exhibit a continuous rather than a discrete change in gene expression.[18,19] We reasoned that the loading of some topics may capture the trajectory along possible state transitions. Indeed, we identified evidence that topic k1 loadings represent the differentiation trajectory from naive to memory B cells. This interpretation is supported by the progressive enrichment of non-naïve B cell populations along the k1 loading axis (Figure 2D, top).

To further validate this biological interpretation, we adapted the ArchR[20] getTrajectory algorithm to analyze GA changes along this putative differentiation path. This showed a coordinated decreased GA for naive B cell marker genes (*IL4R*, *TCL1A*, *IGHM*, and *IGHD*) and increased GA for memory B cell markers (*AIM2*, *CD27*, and *COCH*) (Figure 2D, bottom; Tables S5A–S5D). These findings demonstrate that our topic modeling approach robustly identifies biologically meaningful cell-state transitions.

Finally, we determined the relevance of each topic in terms of explaining complex trait heritability. We used stratified linkage disequilibrium (LD) score regression (s-LDSC)[9] to calculate heritability enrichment of each topic across 50 GWAS traits (Table S6). As expected, we observed greater $h^2_g$ enrichment in many immune-related diseases and blood phenotypes compared to height, a trait we used as negative control. Notably, we found large $h^2_g$ enrichment for three autoimmune diseases (rheumatoid arthritis [RA], systemic lupus erythematosus [SLE], and multiple sclerosis [MS]) in the lymphoid-related topics (Figure 2E). For SLE, B cell topics (k1 and k11) are the most enriched, consistent with the known role of B cells in SLE etiology.[21,22] Monocyte and myeloid-related GWASs (monocyte percentage, monocyte count, and granulocyte percentage) are enriched in monocyte-related topics (k5, k9, k10, k12, k13, and k15; Figure 2E), consistent with the expectation that monocyte cellular programs causally impact myeloid cell numbers and proportions. Interestingly, we found that one topic, k17—which was found in some CD8 T and non-classical T cells—was significantly enriched for heritability for both hospitalized and critical COVID-19 GWASs (Figure 2E). Thus, our analyses identify cell contexts

and regulatory programs that likely mediate genetic risks for several immune traits.

## Topic-derived cell trajectories identify COVID-19-associated cell-state continuums

While our study was not designed to investigate COVID-19, the significant heritability enrichment in accessible peaks linked to topic k17 led us to explore potential associations with COVID-19 status. We assessed whether donor infection status (active COVID-19 vs. controls) explained variation in k17 loadings across cells. Remarkably, k17—which primarily captures CD8 effector memory T (TEM) cells—showed significant enrichment in COVID-19 patients (likelihood-ratio test *p* value = 0.023). Cells with high k17 loadings were disproportionately derived from COVID-19 donors (Figure 2F, top), consistent with an expanded CD8 TEM cell population during infection (Figure S3E).

To further investigate this, we constructed a trajectory based on k17 loadings and stratified cells into loading quintiles (Figure 2F). Differential chromatin accessibility analysis along this trajectory revealed significant changes at loci containing COVID-19-associated genes (Table S7). Notably, we observed progressive chromatin closing at the *PRG4* gene body,[23] coupled with increased accessibility at *CCL3* and *CCR2*[24] loci (Figure 2G).

These findings, combined with the independent observation of COVID-19 heritability enrichment in high-scoring k17 peaks (Figure 2E), suggest that peripheral CD8 TEM cell expansion represents both a cellular signature of COVID-19 and a potential mediator of genetic risk. Our analysis also identified a monocyte-related topic (k15) associated with COVID-19 status (Figure S3F); however, the absence of COVID-19 heritability enrichment in this population suggests that it may represent secondary effects rather than primary genetic risk mediators.

Finally, we compared GA scores between infected and healthy donors across trajectory quintiles. Cells from COVID-19 patients in higher quintiles exhibited substantially more differentially active genes compared to healthy controls (Figure S3G and Table S8). Pathway analysis of these genes revealed significant enrichment in immune-relevant biological processes, including chemotaxis, immune-cell migration, and T cell activation (Figure S3H), consistent with the expected antiviral response to SARS-CoV-2 infection.

These results complement our earlier findings that (1) topic k17 is preferentially enriched in cells from COVID-19 patients, and (2) k17-associated chromatin regions show significant COVID-19 heritability. This continuous spectrum of transcriptional changes was not detectable through conventional cluster-based approaches, highlighting the advantage of a topic-modeling framework in capturing biologically meaningful cell states during infection.

## A HIGH-RESOLUTION MAP OF caQTLs IN PBMCs

We next used our harmonized dataset to map the impact of genetic variation on chromatin accessibility in multiple cell types. To map caQTLs, we first used RASQUAL[25] to model both intra-individual allelic imbalance and interindividual variation in

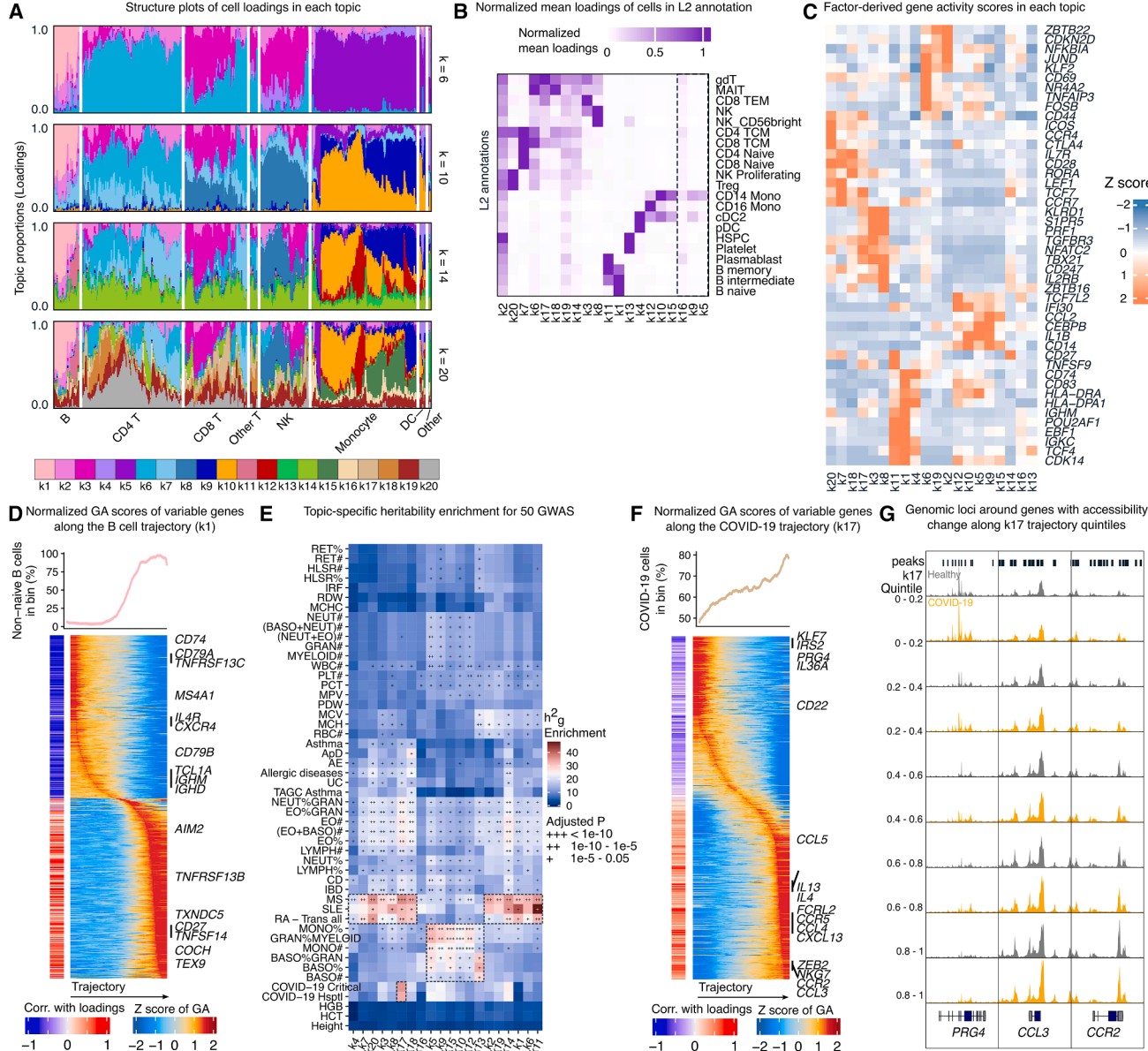

**Figure 2. Topic modeling helps interpretation of intercellular and interindividual variation in scATAC-seq profiles**

(A) STRUCTURE plots of topic loadings in 2,000 randomly selected cells when fitting 6, 10, 14, and 20 topics. Cells were grouped by seven common immune-cell types to highlight coarse-grained differences in topic loading among cell types.

(B) Heatmap showing the average loading for each topic in each cell type in L2 annotation. Dashed box indicates topics without clear enrichment in any cell types.

(C) Heatmap of Z-score-normalized gene-level scores calculated from peak-level scores in each topic.

(D) Top: smoothed percentage of non-naive B cells along the trajectory percentile. Bottom: heatmap on the left shows the Spearman correlation between gene-activity (GA) scores and memory B cell trajectory; heatmap on the right shows row-normalized GA score changing along the trajectory.

(E) Heritability enrichment of 50 GWASs using peaks with the highest 10% of scores in each topic. Dashed boxes highlight specific enrichment results discussed in the main text.

(F) Similar to (D), showing the trajectory and relevant genes in COVID-19-associated topic k17.

(G) Genome browser tracks of the genomic region around three genes (*PRG4*, *CCL3*, and *CCR2*) that progressively gained or lost accessibility along the k17 trajectory. Cells are grouped by disease status and k17 quintiles.

chromatin accessibility for SNPs in a 10 kb window flanking the peak center in PBMCs (aggregating all cells from a donor) as well as the seven immune-cell types defined in our L1 annotation.

Based on our observations that COVID-19 and healthy samples exhibited minimal differences in both cell-clustering patterns and topic-model distributions (with only k17 showing significant association), we performed caQTL mapping using a

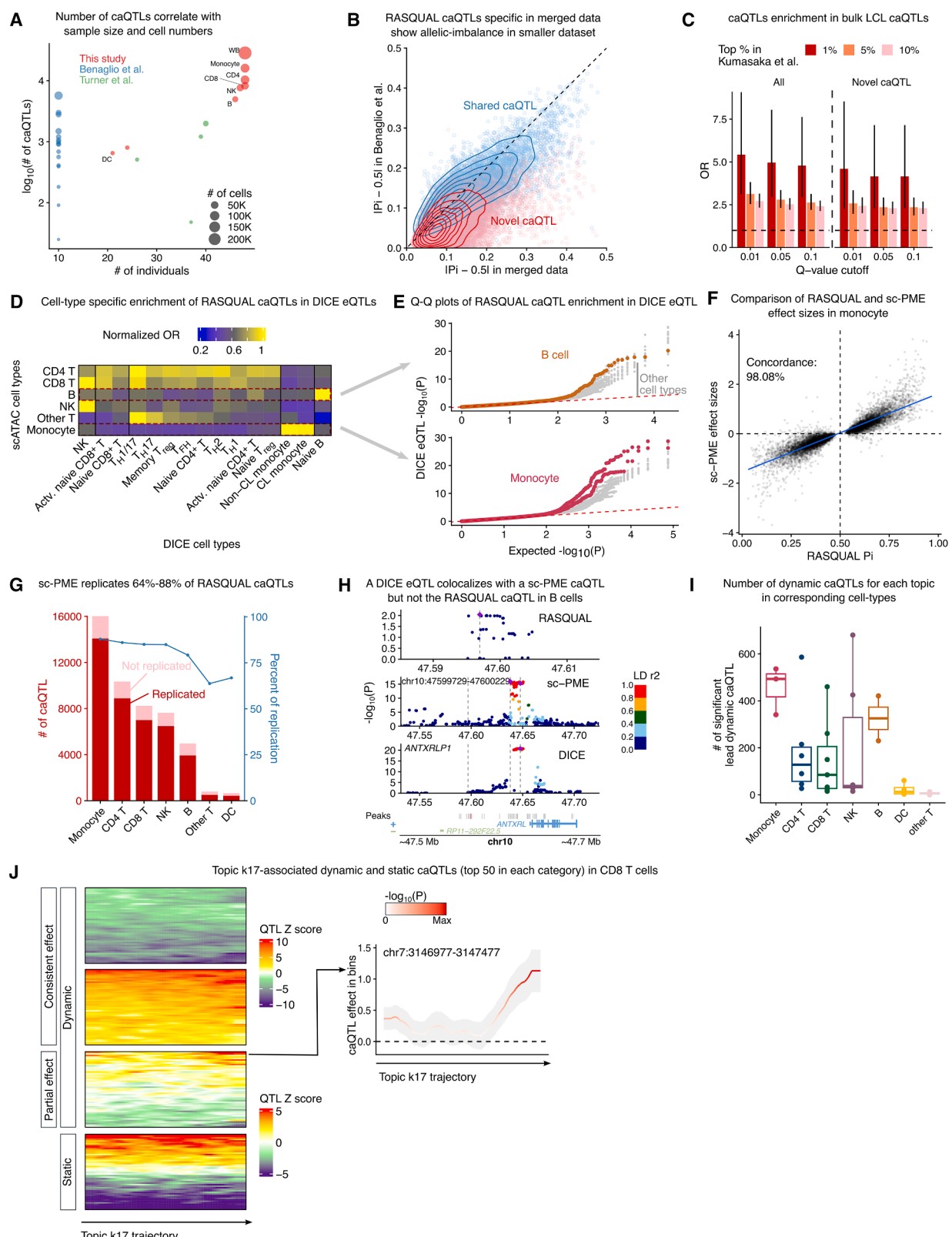

A. Number of caQTLs correlate with sample size and cell numbers

B. RASQUAL caQTLs specific in merged data show allelic-imbalance in smaller dataset

C. caQTLs enrichment in bulk LCL caQTLs

D. Cell-type specific enrichment of RASQUAL caQTLs in DICE eQTLs

E. Q-Q plots of RASQUAL caQTL enrichment in DICE eQTL

F. Comparison of RASQUAL and sc-PME effect sizes in monocyte

G. sc-PME replicates 64%-88% of RASQUAL caQTLs

H. A DICE eQTL colocalizes with a sc-PME caQTL but not the RASQUAL caQTL in B cells

I. Number of dynamic caQTLs for each topic in corresponding cell-types

J. Topic k17-associated dynamic and static caQTLs (top 50 in each category) in CD8 T cells

*(legend on next page)*

combined dataset without separating cells by disease status. We used phenotypic principal components (PCs), genotype PCs, and multiple quality control measurements as covariates[26] (Figure S6A). In total, we identified 37,390 caQTLs (corresponding peaks are referred as cPeaks hereafter), 8,792 of which were discovered in L1 cell types, but not when aggregating all PBMCs (Figure 3A). To verify that the caQTLs we found are likely true positives, we separately mapped PBMC caQTLs in each of the three scATAC-seq datasets. We found multiple lines of evidence indicating that the caQTLs uniquely identified in our harmonized data ("novel caQTLs") are bona fide caQTLs. First, novel caQTLs showed allelic imbalance in the smaller dataset from Benaglio et al.,[10] despite being non-significant (Figure 3B). Second, both our total caQTL set and novel caQTLs are strongly enriched in bulk caQTLs from lymphoblastoid cell lines (LCLs)[27] (Figure 3C). Finally, caQTLs in L1 cell types showed cell-type-specific enrichment in eQTLs from 15 immune-cell types in the DICE consortium.[1,28] For instance, caQTLs in monocytes and B cells were the most enriched for eQTLs in classical/non-classical monocytes and naive B cells in DICE, respectively (Figure 3D). Similarly, caQTLs in CD4 T cells were broadly enriched in eQTLs across various T cell subtypes in DICE. When visualizing the $p$ value distribution of caQTL SNPs in DICE eQTLs, a similar trend emerged (Figure 3E). A major goal of our work is to determine the extent to which caQTLs can explain immune-disease-associated loci. However, RASQUAL does not report effect size and standard error of identified caQTLs, which complicates statistical integration of RASQUAL caQTLs with GWAS summary statistics. Additionally, caQTLs mapped using RASQUAL are biased toward heterozygous SNPs with high read coverage, as SNPs outside peaks have weaker or no allele-specific signal and lower phasing accuracy. To adapt single-cell caQTL data for colocalization,[29] transcriptome-wide association study,[30,31] mashr,[32] and meta-analysis,[33] we used the single-cell Poisson mixed-effects model (sc-PME)[5] to generate standard summary statistics of caQTL effects. In addition to providing critical statistics for downstream statistical analyses, sc-PME allows a larger mapping window (250 kb) compared to the 10 kb used in RASQUAL. This window is difficult to extend in RASQUAL because it requires haplotype phasing, which worsens as a function of distance. Yet, a larger mapping window may help capture additional significant caQTLs (Figure S6C).

To reduce computational time and avoid inflated $p$ values due to high dropout rates in scATAC count data, we restricted our analysis to the 37,390 significant cPeaks from RASQUAL in PBMCs and L1 cell types. As expected, rare cell populations (dendritic cells [DCs] and "other T cells") yielded fewer caQTLs (5,006 and 3,662, respectively, at 10% false discovery rate [FDR]), while more abundant cell types showed substantially higher QTL yields (ranging from 10,739 to 22,785 caQTLs, with a mean of 16,297 at 10% FDR).

We then compared sc-PME results to those from RASQUAL, focusing specifically on the effect-size estimates at top caQTL SNPs. This revealed strong concordance, with an average 79.1% replication rate of caQTLs across cell types (lowest in rare cell types) and high agreement in effect sizes (98.08% in monocytes; Figures 3F and 3G). sc-PME outperformed other single-cell models and yielded results consistent with pseudo-bulk QTLtools[34] (Figure S6D), confirming its robustness (STAR Methods; Figures S7 and S8). Importantly, sc-PME's larger mapping window identified distal caQTLs inaccessible to RASQUAL. Analysis of 1,337 B cell peaks with lead sc-PME SNPs >10 kb away revealed significant enrichment for true caQTLs (8.9% vs. 3.4% genome-wide, $p < 2e{-}16$). A clear example is a cPeak where the sc-PME caQTL is 50 kb upstream and colocalizes with an eQTL (posterior probability 4 [PP4] = 0.95), unlike the local RASQUAL signal (Figure 3H). These distal caQTLs often affect co-accessible peaks (STAR Methods and Figure S9), providing mechanistic insight. Beyond supplying crucial summary statistics, sc-PME uniquely detects distal causal SNPs.

In addition to identifying caQTLs in the L1 cell types, we also discovered dynamic genetic effects along trajectories identified by topic modeling. As shown earlier, these loadings can represent continuous cell states across different axes of biological variation. To detect dynamic caQTLs, we tested for linear interactions between lead caQTLs in L1 cell types and each topic's

---

**Figure 3. Mapping of caQTL with RASQUAL and sc-PME model**

(A) Scatterplot comparing the number of RASQUAL caQTLs as a function of cell number and sample size in this study and two other published scATAC-seq caQTL studies.

(B) RASQUAL effect sizes in our merged data and Benaglio et al. data are highly correlated. Novel caQTLs found in merged data show allelic imbalance in Benaglio et al.,[10] albeit with smaller effect sizes.

(C) RASQUAL caQTLs in aggregated PBMCs are enriched in bulk caQTLs from LCLs for all significant ones (left) and only novel ones found in our data (right). Error bars represent 95% confidence intervals.

(D) Heatmap showing cell-type-specific enrichment of caQTLs in DICE eQTLs. Odd ratios are normalized by the maximum value in each row.

(E) QQ-plot of RASQUAL caQTLs in DICE eQTLs. Top: caQTLs in B cells show elevated signals only in eQTLs from B cells in DICE; bottom: caQTLs in monocytes show elevated signals only in eQTLs from classical and non-classical monocytes in DICE. All other DICE cell types are colored gray.

(F) RASQUAL and sc-PME caQTLs have highly correlated and concordant effect sizes. Results from monocytes are used as an example here.

(G) Replication of RASQUAL caQTL in sc-PME model in seven common immune-cell types. Barplot shows the number of RASQUAL caQTLs that are replicated or not in sc-PME model; line chart shows the percentage of RASQUAL caQTLs replicated.

(H) An example of a DICE eQTL to gene *ANTXRLP1* in B cells colocalizing with sc-PME caQTL but different from the RASQUAL lead SNP. Vertical dashed lines highlight the genomic coordinates of lead SNPs in RASQUAL, sc-PME, and DICE. The shaded region highlights the mapping window of RASQUAL and its position relative to the mapping window of sc-PME. SNPs are colored by LD to the lead SNP.

(I) Number of dynamic caQTLs in each cell type along relevant trajectories defined by topic loadings. Cell types on the $x$ axis are ordered from the most to the least number of cells.

(J) Left: $Z$ scores of dynamic and non-dynamic caQTLs from CD8 T cells in rolling windows along topic k17 trajectory. Dynamic caQTLs were further categorized by whether they are consistently active along the k17 trajectory or only in part of the trajectory. The most significant 50 caQTLs from each category were plotted. Right: effect sizes of one representative dynamic caQTL.

**CellPress**

**Cell Genomics**
Article

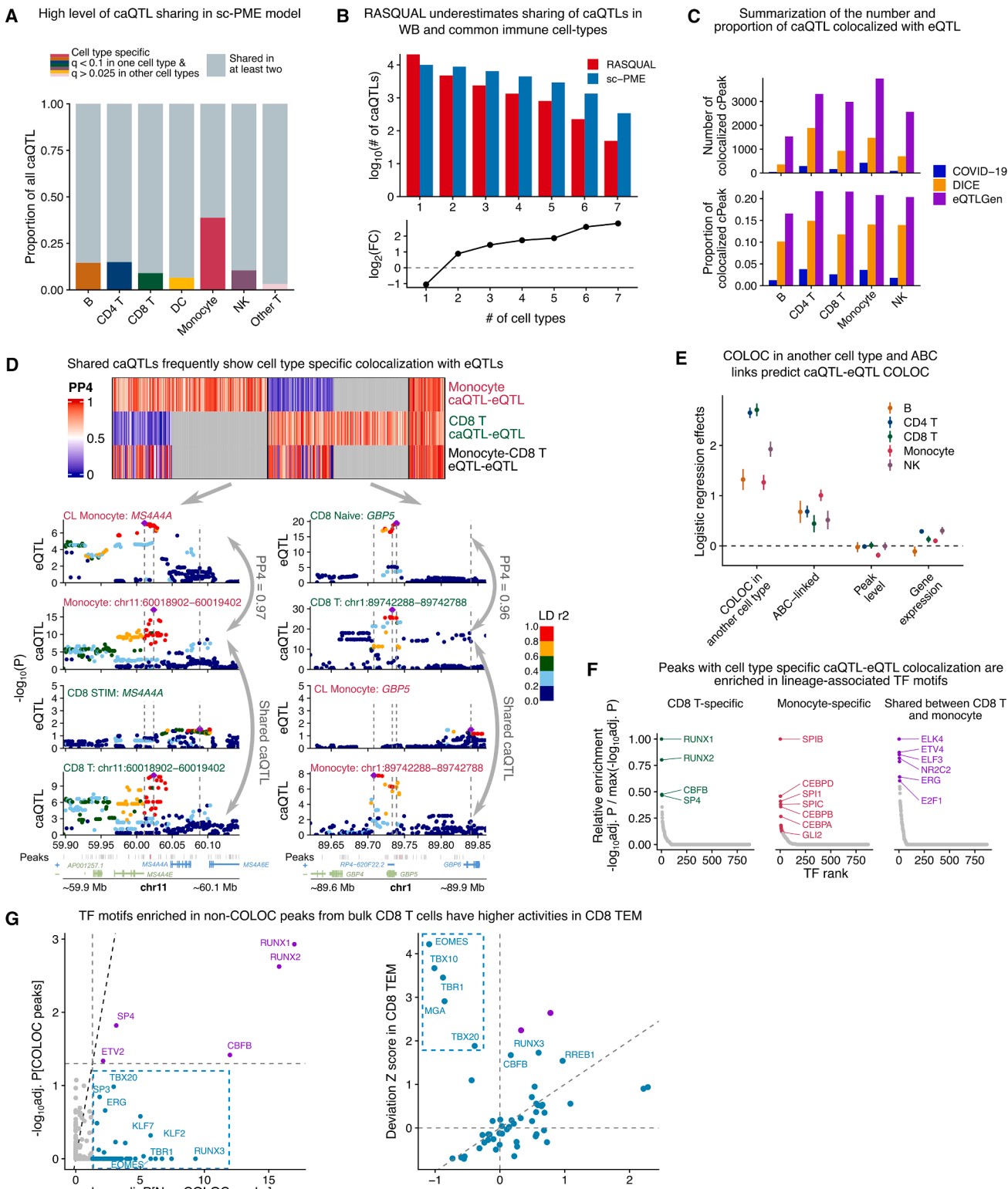

**A** High level of caQTL sharing in sc-PME model

**B** RASQUAL underestimates sharing of caQTLs in WB and common immune cell-types

**C** Summarization of the number and proportion of caQTL colocalized with eQTL

**D** Shared caQTLs frequently show cell type specific colocalization with eQTLs

**E** COLOC in another cell type and ABC links predict caQTL-eQTL COLOC

**F** Peaks with cell type specific caQTL-eQTL colocalization are enriched in lineage-associated TF motifs

**G** TF motifs enriched in non-COLOC peaks from bulk CD8 T cells have higher activities in CD8 TEM

**Figure 4. caQTL-eQTL colocalization and dynamic caQTL mapping**
(A) Sharing of caQTLs in sc-PME model in five common immune-cell types.
(B) Comparison of caQTL sharing in seven immune-cell types for RASQUAL and sc-PME model. Top: barplot of number of caQTLs shared in a given number of contexts. Bottom: log$_2$ fold change in the number of caQTLs between sc-PME and RASQUAL, highlighting the increased level of sharing for sc-PME caQTLs.

*(legend continued on next page)*

loadings (STAR Methods) while restricting our analysis within individual cell types to avoid confounding with cell-type-specific caQTLs. In total, we found 4,200 peaks (q value < 0.01) harboring at least one dynamic caQTL across cell-type-topic combinations. On average, we identified 159 dynamic caQTLs for each cell-type-topic pair, with the number of caQTLs strongly influenced by the number of cells in a given topic (Figure 3I).

As an illustrative example, we examined k17-interacting caQTLs in CD8 T cells, categorizing them into two groups: consistently significant caQTLs, i.e., significant throughout the k17 trajectory but exhibiting varying effect sizes; and partially significant caQTLs, i.e., showing effects only in specific trajectory segments. We visualized how these dynamic caQTL effect sizes change along the k17 trajectory compared to non-dynamic caQTLs (LRT $p > 0.5$) (Figure 3J). Using the activity-by-contact (ABC) model[35] to link these dynamic peaks to genes, we observed enrichment in immune- and disease-related pathways,[36] such as NK-cell-mediated cytotoxicity (GO:0002228, $p = 2.61 \times 10^{-8}$) and regulation of lymphocyte activation (GO:0051249, $p = 3.36 \times 10^{-7}$) (Figure S6E). Notably, several ABC-linked genes overlapped with k17-associated COVID-19 genes from our topic analysis, including *IFNLR1*, *IL5RA*, *IL6ST*, *KLRC4*, *KLRD1*, *KLRK1*, *SYNGR1*, *TLR1*, and *TNFSF14*.

In summary, we established a map of 37,390 static and 4,200 dynamic caQTLs in PBMCs and common immune-cell types. These caQTLs capture the impact of genetic variants on chromatin accessibility in steady state as well as dynamic effects that manifest in important cell contexts such as cytotoxic cells. Summary statistics for all caQTLs are available on Zenodo (https://doi.org/10.5281/ZENODO.14652965).

### Sharing and specificity of caQTLs and eQTLs across cell types and states

We systematically analyzed the sharing of caQTL effects across seven immune-cell types using sc-PME caQTLs. Contrary to some single-cell studies[10,37,38] but consistent with bulk analyses,[1,39–41] we found that most caQTLs are shared across cell types; only monocyte caQTLs showed a high degree of specificity (42.5%). Excluding monocyte-specific effects, only 12.9% of caQTLs showed cell-type specificity on average (Figure 4A). Closely related cell types shared substantially more cPeaks and lead caQTL SNPs (STAR Methods and Figure S10). Interestingly, these results suggest much higher QTL sharing than those obtained from RASQUAL caQTLs (Figure 4B). Thus, these discrepancies are likely explained by statistical power, and further increasing QTL mapping power should yield even higher estimates of sharing. For this reason, we used sc-PME caQTL results for all analyses below.

Since caQTLs are expected to regulate promoter or enhancer activity and consequently influence expression of nearby genes, we investigated whether our sc-PME caQTLs coincided with eQTLs from the DICE dataset[1,28] or from >30,000 whole blood samples in the eQTLGen consortium.[42] The DICE dataset, comprising 15 bulk-sorted immune-cell types from approximately 90 individuals, allowed us to match five cell types from our scATAC data (excluding DCs and "other T cells" in the L1 annotation; Figure S11A). Our analysis revealed 6,228 unique caQTL-eQTL pairs colocalizing in at least one DICE-matched context (termed COLOC caQTL-eQTL pairs), encompassing 2,635 eGenes (24% of all tested; Table S9). The most colocalization signals found was with eQTLGen whole blood eQTLs (9,346 caQTLs, 32.4%), attributable to the enhanced statistical power of this large dataset. Nevertheless, we prioritized DICE-based colocalizations for downstream analyses, as they provide cell-type-specific resolution.

Our analysis of eQTL-caQTL colocalization patterns revealed two key findings. First, only a minority of caQTLs (4,088 cPeaks, representing 18.6% of all tested) showed colocalization with eQTLs across the five examined contexts, indicating that most caQTLs do not appear to influence expression levels of nearby genes in the immune-cell types surveyed by DICE (Figure 4C). Second, when colocalization did occur, it was typically restricted to a single cellular context: 84.5% (5,995) of all caQTL-eQTL pairs were observed in just one context. This striking context specificity contrasts sharply with the broad sharing patterns we observed for caQTLs themselves.

The difference in caQTL sharing and eQTL colocalization was particularly evident in our comparison of CD8 T cells and monocytes. While 3,091 caQTLs showed effects in both cell types, only 134 colocalized with the same eQTL in both contexts, despite the identification of 4,805 and 7,425 eQTL-caQTL pairs in CD8 T cells and monocytes, respectively. This pattern can be seen at specific loci: the shared cPeak at chr11:60018902–60019402 showed colocalization with an *MS4A4A* eQTL only in monocytes, while the shared chr1:89742288–89742788 cPeak colocalized with an *GBP5* eQTL exclusively in naive CD8 T cells (Figure 4D). In both cases, the absence of colocalization in the other cell type reflected a genuine lack of eQTL signals rather than lack of statistical power to detect an eQTL effect. We did not find significant differences in genomic annotation or effect sizes between caQTLs that colocalize with eQTLs and those that do not colocalize (STAR Methods and Figure S12). These observations demonstrate that eQTL effects exhibit substantially greater cell-type specificity than caQTLs across numerous genomic loci.

To investigate why caQTLs influence gene expression in a cell-type-specific manner despite affecting chromatin accessibility

(C) Barplot for the number (top) and proportion (bottom) of caQTLs that colocalize with eQTLs from DICE, our in-sample COVID-19 eQTL, and eQTLGen.

(D) Top: heatmap for PP4 of eQTL-caQTL COLOC for shared caQTLs between CD8 T cells and monocytes, and COLOC of eQTLs in CD8 T and monocytes, in DICE. Bottom: example LocusZoom plots for shared caQTL colocalizing with cell-type-specific eQTLs in monocytes and CD8 T cells, respectively.

(E) Logistic regression coefficients and 95% confidence intervals for variables that predict caQTL-eQTL colocalization. Model includes all caQTL-eQTL pairs that are tested for colocalization.

(F) TF motif enrichment in peaks whose caQTLs colocalize with eQTLs in CD8 T cells and monocytes specifically or in both cell types. Colored points represent TFs with adjusted enrichment $p$ values <0.05.

(G) TF motifs enriched in non-COLOC peaks in CD8 T cells specifically show higher-deviation $Z$ scores in CD8 effector memory T (TEM) cells in our scATAC data.

across multiple cell types, we identified genomic features predictive of caQTL-eQTL colocalization. Using logistic regression, we assessed four factors: (1) gene-expression levels, (2) chromatin accessibility levels, (3) enhancer-to-gene links from the ABC model[35] in matched cell types, and (4) whether the caQTL-eQTL pair colocalizes in multiple cell types.

Neither gene-expression nor chromatin-accessibility levels predicted colocalization, suggesting that statistical power does not explain the limited impact of caQTLs on gene expression. Notably, while many colocalizations were cell-type specific, the strongest predictor was the presence of a colocalization in another cell type (Figure 4E), supporting the idea that cis-regulatory elements—especially promoters—often regulate the same genes across cell types. This was most evident in closely related cell types (e.g., CD4 and CD8 T cells), which shared more caQTL-eQTL colocalizations (Figure S11C). The second most predictive feature was ABC-derived enhancer-to-gene links (strongest in monocytes: $\log_2$ effect = 1.007, $p = 1.63 \times 10^{-66}$; weakest in CD8 T cells: $\log_2$ effect = 0.44, $p = 4.70 \times 10^{-7}$; logistic regression), providing a mechanism for cell-type-specific caQTL effects. These results emphasize the need to differentiate between "merely active" and "functional" accessibility peaks in caQTL analyses. Importantly, our findings suggest that many caQTLs do not affect gene expression in a given cell type simply because they lack physical connections to genes in that context.

We next investigated whether specific TF binding motifs could predict eQTL-caQTL colocalization in a cell type. Comparing CD8 T cells and monocytes—two cell types with well-characterized, distinct TF-driven accessibility patterns—we found that cPeaks with cell-type-specific colocalizations were enriched for lineage-defining TF motifs. For example, CD8 T cell-specific colocalizations were more likely to be associated with RUNX1, RUNX2, and CBFB motifs, while monocyte-specific colocalizations were linked to CEBPA, CEBPB, and SPIB. In contrast, shared colocalizations were enriched for broadly active TFs such as ELK4 and E2F1 (Figure 4F), suggesting that cell-type-specific TFs potentiate the impact of caQTLs on gene expression in a cell-type-specific manner.

To assess the regulatory potential of cPeaks without eQTL colocalization, we analyzed their TF motif enrichments. In CD8 T cells, motifs for EOMES, RUNX3, and TBX20 were uniquely enriched in non-colocalizing cPeaks (Figure 4G, left). Consistent with the known function of EOMES and RUNX factors in TEM functions,[43–45] these TFs are enriched within accessible regions in CD8 TEM cells (Figure 4G, right), suggesting that these caQTLs may regulate gene expression in specific TEM states not captured by the DICE eQTL dataset, such as exhaustion or long-term memory.

In summary, while genetic variants affecting chromatin accessibility are detectable across many cell types, their impact on gene expression depends on the presence of lineage-defining TFs. This explains why eQTL effects are often cell-type specific and are challenging to infer from caQTLs alone.

## GWAS loci colocalize more often with caQTLs than eQTLs

We next assessed the functional relevance of our caQTLs by performing colocalization analyses with immune-related GWAS loci, including 11 immune-mediated diseases, two COVID-19 phenotypes, and 36 blood traits. Comparing these results with eQTL-GWAS colocalizations from the DICE project[28] revealed that 56.8% (4,696 of 8,271) of GWAS loci colocalized with either caQTLs (2,015 loci), eQTLs (1,149 loci), or both (1,532 loci) (Tables S10 and S11).

Among GWAS loci that colocalize with both caQTLs and eQTLs, we found an RA-associated variant[46] near PVRIG (also known as CD112R), which colocalized with a caQTL in CD4 T cells (chr7:99818835–99819335, PP4 = 0.96) and eQTLs in T follicular helper cells (PP4 = 0.94) and naive Treg cells (PP4 = 0.93). The convergence of chromatin accessibility and gene-expression signals suggests that genetic regulation of PVRIG, a T cell co-inhibitory receptor, contributes to RA risk through modulation of T cell activation (Figure S11D).

In contrast, some GWAS loci only colocalized with eQTLs, revealing potentially distinct regulatory mechanisms. An example was the RPS26 locus associated with RA (rs11171739), which showed strong eQTL colocalization across all immune-cell types (PP4 = 0.93–0.95) but no significant caQTL colocalization (PP4 = 0.19 in B cells). While the RPS26 promoter was accessible in all cell types, the lack of caQTL colocalization suggests that post-transcriptional processes[47] may mediate its association with disease (Figure S11E).

Interpreting GWAS loci that colocalize with caQTLs but not eQTLs (caQTL-only) is more challenging. While these variants clearly influence chromatin accessibility, their lack of effect on gene expression in the studied cell types makes it difficult to identify their target genes or relevant cellular contexts. These loci may represent poised regulatory elements that require additional cellular stimuli to exert their effects on transcription. Nevertheless, this category represents the largest fraction of colocalized GWAS loci, expanding colocalized loci from 32.9% (using eQTLs alone) to 57.3% when incorporating caQTL data (Figure 5A). This observation aligns with emerging evidence from recent studies,[6,7,48] demonstrating that many disease-associated variants affect chromatin-level phenotypes without detectable changes in steady-state mRNA levels. The prevalence of such caQTL-only associations raises important questions about their potential mechanisms of action.

In total, 220 (15.3%) caQTL-only loci mapped to promoter regions, which allowed us to nominate a putative causal gene. For example, we identified a peak ~100 bp upstream of ZFP36L1. In CD4 T cells, the same caQTL colocalized with multiple GWASs including Crohn's disease, ulcerative colitis (UC), RA, and MS (PP4 = 0.61–0.84), suggesting that it may affect a common mechanism underlying multiple autoimmune diseases. ZFP36L1 is an RNA-binding protein that is differentially expressed in osteoarthritis and celiac disease and has been shown to regulate T cell and B cell development,[49–51] although its exact mechanism in other autoimmune diseases remains unknown (Figure S11F). In another case, the promoter peak (chr22:37256806–37257306) for gene NCF4 has a caQTL colocalizing with an RA GWAS locus in NK and CD8 T cells (PP4 = 0.78–0.80, Figure S11G). NCF4 has been identified previously as a risk gene for RA by genetic associations; it might be linked to NADPH metabolism in RA and can also regulate NK/CD8 T cell frequencies.[52–54] However, previous eQTL studies did not nominate this gene for RA. These cases

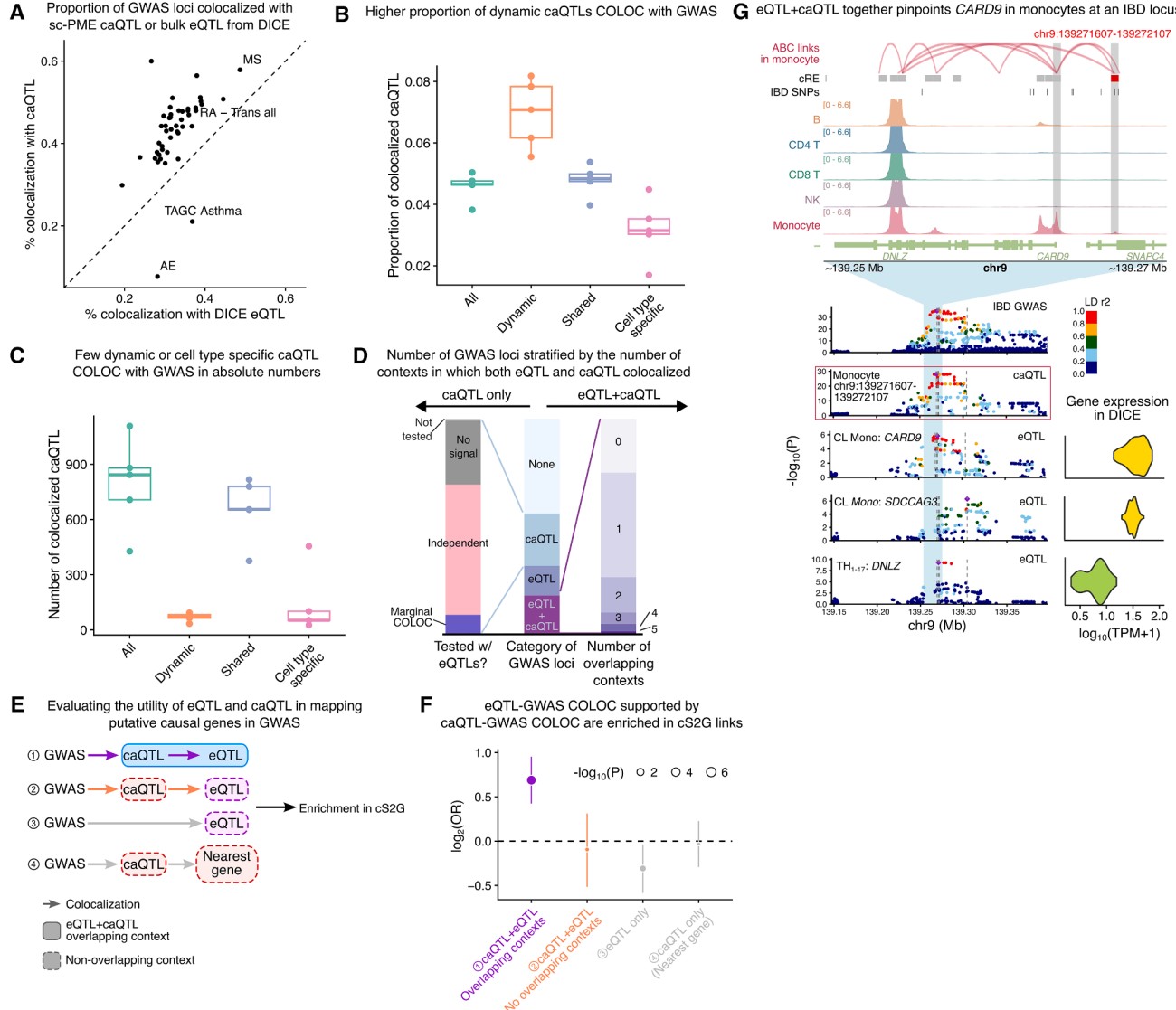

**Figure 5. Widespread sharing of caQTLs impedes functional interpretation of disease GWAS**

(A) Scatterplot for the percentage of colocalized GWAS loci with eQTLs in DICE and caQTLs.

(B) The proportion of cPeaks colocalizing with GWAS, categorized by all peaks, dynamic peaks, cell-type-specific peaks, and shared peaks in each of the five cell types.

(C) The number of cPeaks colocalizing with GWAS, categorized by all peaks, dynamic peaks, cell-type-specific peaks, and shared peaks in each of the five cell types.

(D) Characterization of colocalized GWAS loci by the number of contexts in which they colocalize with either eQTLs, caQTLs, or both.

(E) Schematic representation for testing the utility of eQTLs and caQTLs in mapping putatively causal genes for GWAS. GWAS loci were grouped by their colocalization with either eQTLs, caQTLs, or both, in the same cell-type context or not. GWAS loci that only have caQTLs are mapped to the nearest gene.

(F) Forest plot showing that restricting eQTL-GWAS pairs to contexts also supported by caQTL-GWAS COLOC increases enrichment in causal S2G (cS2G) links. Error bars represent 95% confidence intervals of log₂(OR) estimates.

(G) Example showing that eQTL- and caQTL-GWAS COLOC together narrows down *CARD9* gene in monocytes as the causal gene and context for an IBD locus. Top: genome browser tracks showing chromatin accessibility around *CARD9* locus. Bottom: Manhattan plots for IBD GWASs, colocalized caQTLs, and eQTLs in various cell types (see also Figure S14). Dashed lines highlight the position of lead SNPs in GWASs and the QTL data. Colored box highlights the nominated causal eQTL.

suggest that caQTL-GWAS colocalization can, in a limited number of cases, find putative disease genes. The 220 caQTL-only colocalization gene promoters can be found in Table S12.

We next investigated whether GWAS associations could be more often explained by cell-type-specific caQTLs or by dynamic caQTLs operating along our defined cellular trajectories. Our analysis revealed a notable difference: while cell-type-specific caQTLs showed reduced colocalization rates with GWAS hits compared to shared or all caQTLs combined, dynamic caQTLs demonstrated significantly higher

enrichment for GWAS associations. This pattern suggests that regulatory variants with dynamic effects across cellular states are particularly likely to influence disease-relevant pathways (Figure 5B).

A striking example was for an RA GWAS, where 34.8% (24 of 69) of caQTL-colocalized loci showed dynamic chromatin accessibility effects—a significant enrichment (odds ratio [OR] = 2.27, $p = 1.70 \times 10^{-5}$, Fisher's exact test). This finding implies that tracking regulatory dynamics across cellular transitions may help prioritize functionally relevant GWAS variants. However, the overall contribution of dynamic caQTLs to GWAS colocalizations remains limited, as most caQTLs in our dataset did not exhibit detectable dynamic effects (Figure 5C).

## Limited convergence of caQTL and eQTL signals prevents functional interpretation of most trait-associated loci

Our analyses revealed that most GWAS-caQTL colocalizations lack corresponding eQTL associations. We therefore focused on GWAS loci showing colocalization with both caQTLs and eQTLs (caQTL + eQTL loci), as these offer potential for pinpointing causal disease genes.

To evaluate these loci, we mapped the 15 DICE cell types to our five broader scATAC-seq immune-cell categories (Figure S11A). Surprisingly, we found that 25.4% (375/1,532) of caQTL + eQTL loci showed colocalizations in different cell types (Figure 5D). Furthermore, GWAS loci with multiple context colocalizations typically affected distinct chromatin peaks (cPeaks) and eGenes across these contexts (Figures S13A and S13B). This pattern persisted regardless of whether loci colocalized with caQTLs alone, eQTLs alone, or both, indicating that pleiotropy similarly complicates interpretation of both caQTL and eQTL colocalization analyses. These findings highlight a fundamental challenge in GWAS functional interpretation: the widespread pleiotropy of regulatory variants, which frequently influence distinct genes and chromatin regions across different cellular contexts. We found that 24.4% of caQTL-GWAS colocalizations also colocalize with an eQTL in the same contexts. We hypothesized that eQTLs at these GWAS loci more likely pinpoint the causal disease gene as both caQTL and eQTL converge in the same context. To test this, we leveraged the SNP-to-gene pairs database (cS2G)[55] as a ground truth for causal SNP-to-gene effects. We tested the enrichment of SNP-to-gene links inferred from our QTL data among S2G pairs, using the closest gene as baseline (Figure 5E). We found that caQTL + eQTL S2G pairs in overlapping contexts are significantly enriched for S2G links ($\log_2 OR = 0.70$, $p = 7.64 \times 10^{-8}$; Fisher's exact test), whereas caQTL + eQTL pairs in different contexts were depleted with S2G links ($\log_2 OR = -0.35$, $p = 7.95 \times 10^{-3}$) (Figure 5F). Furthermore, we found no enrichment in S2G pairs when using the eGene to link SNPs to genes at GWAS loci colocalized only with an eQTL ($\log_2 OR = -0.31$, $p = 1.89 \times 10^{-2}$) or when using the nearest TSS to the cPeak at caQTL-only GWAS loci ($\log_2 OR = -0.02$, $p = 0.89$) (Figure 5F). These findings demonstrate that only GWAS loci showing colocalization with both caQTLs and eQTLs in the same cellular context provide meaningful evidence for causal

gene identification. In contrast, loci associated with either caQTLs or eQTLs alone—or in different cell types—offer no greater insight than simply relying on proximity to the nearest gene.

By focusing on GWAS loci with both caQTL and eQTL colocalization in the same cellular context, we prioritized high-confidence causal genes and cell types for hundreds of loci, typically narrowing candidates to $\leq 2$ genes and $\leq 2$ contexts per locus. Many of these prioritized candidates have established roles in disease pathogenesis, as illustrated by an inflammatory bowel disease (IBD) GWAS locus rs3829110 (chr9:139269198). This locus showed colocalization with eQTLs for *CARD9* and *SDCCAG3* in monocytes and *DNLZ* in TH1-17 cells, along with caQTLs in both monocytes and CD4 T cells (Figure S14). Multiple lines of evidence converge to implicate *CARD9* in monocytes as the most likely causal mechanism. First, *CARD9* has well-documented involvement in IBD pathogenesis,[56–58] unlike *SDCCAG3* or *DNLZ*, and represents the only gene linked to this GWAS SNP in the S2G database with the highest possible score (S2G = 1). Second, our analyses revealed that the colocalized chromatin peak (chr9:139271584–139272084) exhibits monocyte-specific accessibility, contains fine-mapped IBD risk variants, and shows ABC-predicted physical connectivity to the *CARD9* promoter exclusively in monocytes (Figure 5G). Third, expression profiling confirmed high *CARD9* expression in monocytes compared to minimal *DNLZ* expression in TH1-17 cells. This example demonstrates both the power of integrative approaches and the limitations of single-modality analyses.

In conclusion, GWAS colocalization with both a caQTL and an eQTL gives us the best chance at identifying the causal genes and contexts underlying trait association. Thus, accurate causal inference from GWAS requires careful integration of multiple complementary datasets rather than simple colocalization with individual molecular phenotypes.

## DISCUSSION

We established a comprehensive scATAC atlas integrated with high-quality genotype data. While single-cell eQTL mapping has become increasingly routine, approaches for single-cell caQTL analysis remain underdeveloped. Our work addresses this gap by demonstrating that combining allelic-imbalance modeling through RASQUAL with our sc-PME framework enables robust identification of high-confidence caQTLs suitable for robust downstream analyses. From our integrated scATAC-seq data, we identified 37,390 caQTLs, a 4-fold increase over previous studies.[10] We also mapped dynamic caQTLs across cellular states using topic modeling, which provided more biologically interpretable trajectories than principal components. This approach captured context-dependent regulatory variation missed by discrete classifications, establishing a robust framework that advances our understanding of genetic regulation at single-cell resolution.

Remarkably, we found that adding caQTLs increases colocalized GWAS loci by an average of ~50% compared to eQTLs alone, suggesting that mapping caQTLs in single cells may be a promising paradigm for studying and biologically interpreting

## Article

disease-associated variants. As such, our work corroborates similar findings reported from recent bulk caQTL studies.[7,59]

Still, there has been no straightforward explanation as to why, in any given cell-type context, many GWAS loci have effects on chromatin but not on gene-expression level. We found that the sharing of caQTLs across immune-cell types is widespread, but their impact on gene-expression levels is much more restricted owing to cell-type-specific gene expression and/or cell-type-specific enhancer-promoter interaction. Thus, we interpret novel caQTL-GWAS colocalization results with caution. We posit that many caQTL-GWAS colocalizations do not reflect meaningful regulatory effects in a causal cell-type context. Thus, finding the gene and cell-type context that causally mediates genetic effects on complex traits may be difficult even when a caQTL-GWAS colocalization has been found. Our findings that caQTLs are widely shared across immune-cell types and states indicate that most of these caQTLs may impact gene-expression levels in immune contexts for which eQTLs are unavailable. Thus, increasing the cell-context coverage maps of eQTLs will help in finding more contexts in which caQTL, eQTL, and GWAS signals all align.

Our results are consistent with poised enhancers, which are accessible in many cell states but functional only upon activation by specific TFs. However, this model is complicated in divergent cell types like B cells and monocytes, where shared open chromatin regions can have distinct regulatory impacts, as seen in RA-associated states where chromatin accessibility is shared but gene expression is not.[60] This prevalence of context-specific regulation complicates the fine-mapping of causal genes from GWAS loci.

In summary, while chromatin accessibility data are valuable for characterizing regulatory elements, many caQTLs without corresponding eQTLs may lack functional consequences. This underscores the critical need to integrate complementary data, such as eQTLs, enhancer-promoter interactions, and other functional evidence, for accurate GWAS interpretation. Our work suggests that future advances will require population-scale multi-omic single-cell studies in disease-relevant contexts.

We remain far from comprehensively identifying the causal cell types for most GWAS associations. Mapping regulatory QTLs in relevant populations—such as cells from patients[1,61] or treated organoids[62,63]—will be necessary to uncover the causal cellular contexts of disease variants.

## Limitations of the study

Our study has several limitations. First, the modest sample size and cell numbers reduce power to detect caQTLs in rare but potentially important cell subtypes in disease. Second, due to the computational demands of the sc-PME model, our caQTL mapping was restricted to the cPeaks identified using RASQUAL. More efficient and statistically robust methods will be necessary for comprehensive genome-wide discovery of caQTLs and their dynamic effects at single-cell resolution.

## RESOURCE AVAILABILITY

### Lead contact

Requests for resources and reagents should be directed to and will be fulfilled by the lead contact, Yang I. Li (yangili1@uchicago.edu).

### Materials availability

This study did not generate new unique reagents.

### Data and code availability

- The code generated during this study is available at GitHub (https://github.com/Zepeng-Mu/COVID-19_scATAC_caQTL_manuscript) and Zenodo[64] (https://doi.org/10.5281/zenodo.17049487).
- Raw fastq files for scATAC data collected in this paper are available in the Sequence Read Archive (PRJNA1322323).
- Full caQTL summary statistics from sc-PME model can be found on Zenodo[65] (https://doi.org/10.5281/zenodo.14652965).

## ACKNOWLEDGMENTS

We thank N. Gonzales for her careful reading of our manuscript and insightful comments. This work was completed in part with resources provided by the University of Chicago Research Computing Center. This work was supported by National Institutes of Health grants R35GM153249 (Y.I.L.), R01GM130738 (Z.M. and Y.I.L.), R01GM134376 (L.B.B.), and R35GM152227 (L.B.B.). This work was also supported by the Chan Zuckerberg Biohub Chicago (L.B.B. and Y.I.L.). This work was also supported by Canadian Institutes of Health Research grants VR2-173203 and 178344 to D.E.K. We also acknowledge support from the UChicago DDRCC, Center for Interdisciplinary Study of Inflammatory Intestinal Disorders (C-IID) (NIDDK P30 DK042086). H.E.R. was supported by a Ruth L. Kirschstein National Research Service award (NHLBI F31-HL156419). The Biobanque Québécoise de la COVID-19 (BQC19) is supported by the FRQS Génome Québec and the Public Health Agency of Canada.

## AUTHOR CONTRIBUTIONS

Y.I.L. and L.B.B. jointly supervised research; Z.M., X.L., Y.I.L., and L.B.B. conceived and designed the experiments; H.E.R., E.K., A.D., V.L., C.B., and D.E.K. performed the experiments; Z.M. performed statistical analyses and analyzed the data; H.E.R. and R.A.-G. also performed data analysis; and Z.M. and Y.I.L. wrote the paper, with critical contributions from L.B.B. and X.L. and input from all authors.

## DECLARATION OF INTERESTS

The authors declare no competing interests.

## DECLARATION OF GENERATIVE AI AND AI-ASSISTED TECHNOLOGIES IN THE WRITING PROCESS

The authors used DeepSeek to improve the clarity of the manuscript. The authors reviewed the content and take full responsibility for the content of the publication.

## STAR★METHODS

Detailed methods are provided in the online version of this paper and include the following:

- KEY RESOURCES TABLE
- EXPERIMENTAL MODEL AND STUDY PARTICIPANT DETAILS
  - Clinical sample collection
- METHOD DETAILS
  - Human sample processing
  - scATAC-seq library preparation
- QUANTIFICATION AND STATISTICAL ANALYSIS
  - Preprocessing of in-house and public scATAC-seq data
  - Preprocessing and cell-type annotation of public scRNA-seq data
  - Basic analysis of scATAC-seq data
  - Genotype imputation from aggregated scATAC-seq data
  - Topic modeling on scATAC count data
  - Topic model captures batch effects and technical variations

- o Chromatin accessibility QTL mapping
- o Colocalization of caQTLs with bulk eQTLs and GWAS

**SUPPLEMENTAL INFORMATION**

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

# STAR★METHODS

## KEY RESOURCES TABLE

| REAGENT or RESOURCE | SOURCE | IDENTIFIER |
| --- | --- | --- |
| **Critical commercial assays** | | |
| 10X Genomics Chromium Next GEM Single Cell ATAC Reagent kit v1.1 | 10X Genomics | PN1000175 |
| 10X Genomics Chromium Next GEM Chip H Single Cell Kit | 10X Genomics | PN1000162 |
| 10X Genomics Single Index Kit N, Set A, 96 reactions | 10X Genomics | PN1000212 |
| **Deposited data** | | |
| scATAC raw fastq files | This study | SRA: PRJNA1322323 |
| scATAC-seq data from healthy donors | Benaglio et al.[10] | GEO: GSE163160, GSE199253 |
| Genotype data from Benaglio et al. | Benaglio et al.[10] | EGA: EGAS000 01006184 |
| scATAC-seq data from COVID-19 convalescent donors | You et al.[11] | SRA: PRJNA718009 |
| Summary statistics for sc-PME caQTL | This study | https://doi.org/10.5281/ZENODO.14652965 |
| COVID-19 scRNA-seq data | Stephenson et al.[12] | https://www.covid19cellatlas.org/ |
| cS2G | Gazal et al.[55] | https://alkesgroup.broadinstitute.org/cS2G |
| Activity-by-Contact (ABC) model | Nasser et al.[35] | https://mitra.stanford.edu/engreitz/oak/public/Nasser2021/AllPredictions.AvgHiC.ABC0.015.minus150.ForABCPaperV3.txt.gz |
| DICE bulk eQTL from our previous publication | Mu et al.[1] | https://doi.org/10.5281/zenodo.4480206 |
| eQTLGen *cis*-eQTL v1.0 summary statistics | Võsa et al.[42] | https://eqtlgen.org/cis-eqtls.html |
| **Software and algorithms** | | |
| GLIMPSE | Rubinacci et al.[66] | https://odelaneau.github.io/GLIMPSE/glimpse1/ |
| fastTopics | Carbonetto et al.[17] | https://stephenslab.github.io/fastTopics/index.html |
| RASQUAL | Kumasaka et al.[25] | https://github.com/natsuhiko/rasqual |
| QTLtools | Delaneau et al.[34] | https://qtltools.github.io/qtltools/ |
| ArchR | Corces et al.[20] | https://www.archrproject.com/ |
| Custom code | This study[64] | https://doi.org/10.5281/zenodo.17049487 |

## EXPERIMENTAL MODEL AND STUDY PARTICIPANT DETAILS

### Clinical sample collection

We collected PBMCs from 20 COVID-19 patients hospitalized in Montréal, Canada with COVID-19 between April 2020 and December 2021 who initially presented with symptomatic, primary infection. All acute phase samples were collected from unvaccinated patients within 20 days of symptom onset. None received plasma transfer therapy. We also prospectively sample a subset of the same patients during convalescent phase COVID-19 (*n* = 11), and healthy controls (*n* = 5). The respective institutional IRBs approved multicentric protocol: MP-02-2020-8929. Written, informed consent was obtained from all participants or, when incapacitated, their legal guardian before enrollment and sample collection. The demographics of these samples can be found in Table S1.

## METHOD DETAILS

### Human sample processing

Cryopreserved samples were thawed and cultured in RPMI 1640 without glutamine (Fisher) supplemented with 10% fetal bovine serum (Corning), 1% L-glutamine (Fisher), and 0.01% gentamicin from 10 mg/mL stock (Fisher) overnight. After incubation, samples were washed with PBS, passed through a 40 μm filter, and manually counted with trypan blue staining by brightfield hemocytometer.

## scATAC-seq library preparation

As in the demonstrated protocol CG000169, Rev. E, from 10X genomics, we lysed each batch of 1 million cells with a IGEPAL 630/digitonin based lysis buffer for 3 min on ice. Nuclei were then washed once and resuspended in 7.5uL of 10X Genomics Nuclei Buffer. Finally, 2.5 μL of nuclei were counted by trypan blue staining on a hemocytometer.

We used 10X Genomics Chromium Next GEM Single Cell ATAC Reagent kit with version #1.1. Nuclei were transposed by isothermal incubation at 37°C and Post Gel Bead-in-Emulsion (GEMs) were generated by the 10X Controller and subjected to PCR as described in the 10X User Guide, and post-incubation products were stored at −20°C until downstream processing. Each sample was captured individually (i.e., without pooling) aiming for 5,000 nuclei.

Post GEM incubation cleanup and sequencing library preparation were performed as described in the Single Cell ATAC Reagent Kits v1.1 User Guide (10X Genomics). Briefly, we cleaned up post-incubation GEMs first with DynaBeads MyOne SILANE beads (ThermoFisher Scientific) and then with SPRIselect reagent (Beckman Coulter). Libraries were constructed by performing sample index PCR (98°C for 45 s, 9 or 10 cycles of 98°C for 20 s, 67°C for 30 s, 72°C for 20 s, and 72°C 1 min) followed by SPRIselect size selection.

Prior to sequencing, all multiplexed single-cell libraries were quantified using the KAPA Library Quantification Kit for Illumina Platforms (Roche) and pooled in an equimolar ratio. Libraries were sequenced by 100 base pairs (read1: 50, i7: 8, i5: 16, read2: 50) on an Illumina NovaSeq 6000.

## QUANTIFICATION AND STATISTICAL ANALYSIS

### Preprocessing of in-house and public scATAC-seq data

We processed data in all three studies from FASTQ files using the following pipeline. Reads were processed using cellranger-atac v2.1.0 with an in-house GRCh37 reference genome generated from scripts provided by 10X Genomics (https://support.10xgenomics.com/single-cell-atac/software/release-notes/references#GRCh38-2020-A-2.0.0). We removed reads that were unmapped, did not have primary alignment, failed platform/vendor quality checks, and had duplicated or Supplemental alignment; we only kept reads that were paired and mapped in proper pairs ('samtools view -f 3 -F 3844'). We then removed allelic-biased reads using the WASP[67] workflow implemented in Hornet (https://github.com/TheFraserLab/Hornet). We converted the resulting BAM file from each library into a fragment file using sinto v0.7.5 (https://timoast.github.io/sinto/index.html) and loaded into an ArchR project separately. We then analyzed each library separately to identify high-quality barcodes and remove doublets. As a first pass, we excluded cell barcodes with fewer than 1,000 and more than 50,000 unique fragments, with a TSS enrichment score lower than six for all libraries and excluded those with high ratios of reads mapping to nucleosomes, mitochondrial genome or ENCODE blacklist regions in a library-specific manner (Figure S1B). We also used AMULET[68] on BAM files to flag and remove potential doublets (AMULET $q$ value <0.1).

### Preprocessing and cell-type annotation of public scRNA-seq data

We re-analyzed previously published PBMC scRNA-seq data from COVID-19 patients.[12] Count matrix was downloaded from Human Cell Atlas webpage and converted to a Seurat object. As a first pass, we ran Azimuth with PBMC reference to annotate all the cells. We compared Azimuth L1 annotation (B, CD4 T, CD8 T, dendritic cells (DCs), monocytes, natural killer cells (NK), other T cells) with original cell type labels provided by the author and only kept cells with consistent labels.

### Basic analysis of scATAC-seq data

Through the processing steps above, we identified a list of barcodes that represent high-quality single cells with individual ID for each library. We then loaded the fragment files containing these barcodes from all libraries to one ArchR project for integrated analysis. Dimension reduction on this full dataset was performed on the binary tile matrix, selecting the top 30,000 variable tiles and outputting 50 reduced dimensions with 'addIterativeLSI' function in ArchR. We then feed this LSI projection to the 'reducedMNN' function in R package 'batchelor' to remove batch effects across libraries.[69] We implemented a wrapper function to add MNN-adjusted dimensions to the ArchR project object, enabling downstream analysis within ArchR framework. Cell clusters were identified with a resolution of 0.8. For visualization, the reducedMNN-adjusted LSI was used to derive a UMAP embedding with 'minDist = 0.8' and 'spread = 1.'

To calculate gene-activity scores (GA scores) from scATAC-seq profiles, we generated an in-house gene reference set from the GENCODE v19 annotation. Basically, we started from all the gene symbols in the full GENCODE annotation and removed those whose 'gene_type' map to one of the following: snRNA, misc_RNA, snoRNA, rRNA, miRNA, pseudogene, polymorphic_pseudogene, IG_V_pseudogene, TR_V_pseudogene, IG_C_pseudogene, TR_J_pseudogene, IG_J_pseudogene, processed_transcript, sense_intronic, 3prime_overlapping_ncrna and sense_overlapping, keeping 32,885 genes on chr1-22 and chrX. We then extracted the transcription start sites (TSS) and exons for these genes and constructed a gene annotation object that was added into our ArchR project. Our custom annotation includes important marker genes that are missed in the default hg19 annotation used by ArchR, such as gene *LINC02446* (also known as *RP11-291B21.2*, Figure 1F), a long non-coding RNA that marks activated CD8 T cells.[70] Using this custom gene annotation, we then calculated the GA score using 'addGeneScoreMatrix' with default parameters in ArchR.

To better annotate cell-types in our scATAC-seq data, we integrated it with our Azimuth-annotated scRNA-seq data and transferred the annotation labels to scATAC-seq cells. We first performed unconstrained integration using the 'addGeneIntegrationMatrix' function in ArchR. We then examined the confusion matrix between cell clusters and annotated cell-types. Several clusters contained mixed cell-types from the reference dataset. Upon further speculation, we found these clusters tend to have higher rates of mitochondrial DNA and lie between well-defined cell-types in the UMAP, suggesting these cells are of lower quality or are potential unremoved doublets. We excluded these cells from the dataset and performed constrained integration by restricting cells within four groups: T/NK cells, monocytes/DCs, B cells and other (platelet and HSPC). After this round of constrained integration, we found several T/NK cell subtypes in L2 have very low cell numbers in scATAC-seq data, we therefore only kept labels with sufficient cell numbers (CD14 Mono, CD16 Mono, NK, NK_CD56bright, NK Proliferating, pDC, cDC2, Platelet, B naive, B intermediate, B memory, Plasmablast, gdT, MAIT, CD4 Naive, CD4 TCM, Treg, CD8 Naive, CD8 TEM, CD8 TCM, HSPC) and performed another iteration of constrained integration. Finally, we re-calculataed the LSI, MNN-adjusted dimensions and the UMAP embedding and Leiden clustering on the remaining cells.

To identify candidate peaks, we first produced pseudo-bulk group coverages in each Leiden cluster and used the three studies as sample labels in 'addGroupCoverages(sampleLabels = "Sample").' We then called reproducible peak set by setting 'reproducibility = 2' in 'addReproduciblePeakSet.' In this way, we were able to identify peaks that are called in at least two of the three studies in our data.

### Genotype imputation from aggregated scATAC-seq data

Read coverage across the genome was visualized using 'plotCoverage' from deepTools,[71] excluding blacklist regions from ENCODE. Genotype likelihood calculation and imputation were performed following GLIMPSE documentation.[66] Briefly, we first inferred genotype likelihoods across all SNPs in the 1000 Genome Project from filtered BAM files from scATAC data with 'bcftools mpileup.' In this step, we only included sites with sequencing depth below 15 ("bcftools view -i 'FORMAT/DP<=15'") to avoid regions with unreasonably high read coverage. Next, we merged genotype likelihood from all individuals from the three studies and performed GLIMPSE genotype imputation jointly. Imputed genotypes were phased with eagle v2.4.1.[72]

To confirm GLIMPSE-imputed genotypes from scATAC-seq reads are of high quality and are not biased by strong allele-specific signals in accessible chromatin regions, we compared Minimac4 imputation from microarray with GLIMPSE results in the 13 individuals with microarray data from Benaglio et al.[10] We first imputed the microarray genotype data from the original study using the pipeline documented in Michigan Imputation Server[73] (https://imputationserver.readthedocs.io/en/latest/pipeline). We used the same reference panel as in our GLIMPSE pipeline. We then calculated mean imputation quality score (INFO score) for SNPs stratified by reference MAF bins.

We also compared microarray genotyped SNPs to the imputed SNPs in scATAC-seq using 'vcf-stats.' Genotype dosages imputed from scATAC reads and genotyping arrays were highly correlated across all reference minor-allele frequency (MAF) bins (>91%, Figure S1G), indicating that imputed genotypes from scATAC-reads are highly accurate and are not biased by allelic imbalance in chromatin accessibility. We also assessed genotype imputation accuracy and observed a low discordance rate of approximately 5% across SNPs. This error rate was consistent with that observed when using microarray-based imputed genotypes (Figure S1H). Our harmonized callset contains 6.75 million high-quality SNPs for caQTL (Figure S1I).

### Topic modeling on scATAC count data
#### Fitting the topic model

Topic modeling was performed using the R package 'fastTopics.'[17] We retrieved the cell-by-peak count matrix from the ArchR object. In practice, we considered two aspects in fitting Poisson NMF to our count data. First, fitting the topic model on the full data is computationally expensive. Second, the NMF problem is non-convex, meaning that each model fit returns slightly different results, making it difficult to compare the output using different parameters even on the same data. To speed up the model fitting process, we randomly down sampled 10,000 cells. For peaks that have zero counts in these 10,000 sampled cells, instead of removing them from the matrix, we further sampled cells where they have non-zero counts. This ensures that we can project the fitted model to the full count matrix. In total, 10,711 cells were used for the initial model fitting.

To make sure we can easily compare multiple model fits on the same data, we first performed NMF using a small number of total topics (k) and then fit NMF with more topics conditioning on the previous model fit. We started by fitting the topic model with k = 6 using the 'fit_topic_model' function, using 100 main iterations and 200 refining iterations ('numiter.main = 100, numiter.refine = 200'). This returned a multinomial topic model fitting, which was then projected to the full count data using the 'predict' function implemented in 'fastTopics.' To fit a model with eight topics (k = 8), we propagated the loading matrix and the factor matrix from k = 6 with two more columns of uniformly distributed values (1/k for loading matrix and 1/[number of peaks] for score matrix). We then applied the fitting steps adapted from the 'fit_topic_model' function. Briefly, the expanded loading matrix and factor matrix were passed into 'init_poisson_nmf' together with the down-sampled count matrix to initialize a new Poisson model. Then, the model was fitted with an EM algorithm for 100 iterations (main fitting) and updated with SCD algorithm for 200 iterations in two consecutive runs of the 'fit_poisson_nmf' function. The fitted Poisson model was converted to a multinomial NMF model with 'poisson2multinom' function. The output of this final step is a cell-by-eight-topic loading matrix. We iterated this process with 10, 12, 14, 16, 18 and 20 topics. This framework allowed us to keep the order of topics constant, making it possible to compare across model fits, while

updating them when more topics are added. To visualize topic modeling results in a Structure plot, we performed PCA on the loading matrix (after centering and scaling) and used the rotated data matrix for K-means clustering (K = 30). To avoid over-plotting, we randomly selected 2.5% of cells from each cluster, resulting in a subset of ~5,500 cells for visualization.

### Topic model captures batch effects and technical variations

In our topic model results, we observed that several topics (k2, k5, k9, k16, k19) are not enriched in any particular cell type. We sought to understand if these topics are capturing cell quality or other technical variations. We first calculated correlations between topic loadings and continuous measurements of batch and data quality, including MTratio, ReadsInTSS and TSSEnrichment etc. Several topics have significant correlations with these QC metrics. We also calculated mean topic loadings in each dataset and found that most topics are not enriched in any specific dataset (Figure S4). This shows that batch effects are captured by certain topics and have minimal impact on other biologically-relevant topics. We believe this is a desirable property for topic modeling because topics that reflect technical variations can be excluded from downstream analysis. Furthermore, we also regressed out batches and QC metrics in all our downstream analysis, similar to previous methods.[74]

### *Calculation of gene-level scores from peak-level scores*

To define a molecular program underlying each topic, we relied on the factor matrix. We selected the top 10% of peaks with the highest score in each topic. To calculate gene-level scores from peak-level scores, we applied ArchR's exponential-weighting strategy to calculate gene-activity scores. Briefly, scores of peaks within the gene body are directly summed up, and scores of peaks up to 5 kb upstream of the gene TSS are weighted by distance-based power-law. We then calculated the *Z* score of each gene across all the topics.

To benchmark these four strategies, we focused topic k7, which clearly associates with naive CD4 T cell states. Our investigation of well-known naive T cell marker genes revealed three key findings across the different peak-to-gene mapping approaches: First, these markers consistently showed higher scores in the naive T cell topic (k7) compared to other topics for all approaches. Second, established marker genes such as *CCR7* and *LEF1* consistently outperformed less specific genes like *CD8A* within each strategy. However, our third and most interesting observation revealed that while these markers performed well in absolute terms, they ranked surprisingly lower in the TSS and nearest TSS peak methods due to numerous other genes achieving even higher scores (Figure S5B).

Motivated by these observations, we performed gene-set over-representation analysis[75] on top 1000 genes with the highest gene score for each peak-to-gene mapping method. Our analysis of the top five enriched gene sets for each method revealed important differences in biological relevance. The exponential-decay scoring method showed significant enrichment for key T cell pathways including "T cell differentiation" and "T cell activation," while methods relying on TSS proximity or nearest peak association were predominantly enriched for mRNA processing pathways (Figure S5C). This pattern held when examining MSigDB immune hallmark gene sets (C7 collection),[76] where our scoring method and the co-accessibility approach consistently identified gene sets with T cell upregulation functions (Figure S5D), confirming their T-cell-specific expression patterns. In contrast, the TSS-based methods produced enrichment of biologically irrelevant gene sets for T cell biology. Together, these results demonstrate that our exponential decay gene scoring method provides substantially more biologically meaningful results than simply linking peaks based on TSS proximity alone. We used ArchR-like gene scores in all downstream analysis.

### *Stratified-LDSC analysis on top peaks in each topic*

For s-LDSC analysis, we selected top 10% peaks with the highest scores from each topic. Each peak was extended to 1,500 bp around its center. We used GWAS summary statistics for 50 phenotypes (11 immune-related diseases and 36 blood cell-type traits, and height as a negative control) (Table S13). GWAS summary statistics were munged by 'munge_sumstats.py.' LD score calculation and s-LDSC analysis were carried out according to LDSC documentation conditioning on default baseline annotations.

### *Association between topic k17 loadings and COVID-19 status*

To test whether k17 loadings are associated with COVID-19 status, we first calculated average donor-level k17 loadings for each sample. We only included cells from healthy controls or active COVID-19 at the time of sample collection and with k17 loading larger than 0.01, as cells below this cutoff largely represent estimation noise. We then fitted two mixed-effects logistic regression models:

$$logit(COVID - 19) \sim Loading_{k17} + MTratio + nFrags + (1|donor) \qquad \text{(Equation 1)}$$

$$logit(COVID - 19) \sim MTratio + nFrags + (1|donor) \qquad \text{(Equation 2)}$$

Equation 1 is the full model and (2) is the null model. We then performed a likelihood ratio test (LRT) with 'anova' function in R to test whether k17 loadings significantly predicts donor COVID-19 status.

### *Trajectory analysis in topic model*

We defined the cell trajectory directly from topic loadings with slight modifications to accommodate the analysis workflow of the ArchR package. As a proof-of-concept, we first scrutinized the B cell trajectory. Since k2 represents naive B cells, its loadings are the highest in naive B cells and decreases in memory B cells and plasmablast. To construct a trajectory that represents B cell maturation, we used the reverse order of k2 loadings, such that the trajectory value increases as naive B cells transit into memory B cells. The trajectory was restricted to L1-annotated B cells, and we set the value of all other cell-types to 'NA,' so that ArchR does not use these cells in the analysis. The trajectory values were then scaled to the range of 0–100 for downstream analysis. To study the change in the proportion of memory B cells or plasmablasts along the trajectory, the trajectory was divided into percentiles,

and we calculated the proportion of non-naive B cells in each percentile according to L1 annotation. When building a trajectory for k9, we removed cells whose k9 loadings are below 0.1, as these likely represent background noise rather than biologically meaningful variations.

After deriving the trajectory values from cell loadings, we added the trajectories to the ArchR project as a metadata column. To visualize the changes of gene-activity scores along the trajectory, we used the 'getTrajectory' function, followed by 'plotTrajectory-yHeatmap' functions with options 'varCutOff = 0.8, returnMatrix = TRUE.' The heatmap was visualized using ComplexHeatmap.[77]

To assess the relevance of k17 trajectory to COVID-19, we first asked in a cluster-based analysis, how many differentially active genes can be found between healthy and COVID-19 cells. To do so, we compared all COVID-19cells with all healthy cells in k17 trajectory regardless of k17 loadings using the 'getMarkerFeatures' function. We then grouped cells into quintiles according to their k17 loadings, where higher quintiles were enriched for more COVID-19 cells. We next tested for differential gene activity between all cells in the first quintiles and COVID-19 cells in the higher quintiles (second and above). Note in this test we used healthy and COVID-19 cells in the first quintile as control, following the idea that COVID-19 cells in the first quintile are epigenetically similar to healthy cells.

### Chromatin accessibility QTL mapping

#### RASQUAL caQTL mapping

Chromatin accessibility QTL (caQTL) were first mapped using RASQUAL on two grouping levels, PBMC-like by aggregating all cells from a sample and L1 annotations. We generated pseudobulk counts by summing single-cell counts across cell barcodes within each group. Because a subset of our COVID-19 samples are collected from the same individuals at two visits, we only included the time point with more cells to avoid repeated measurement in the RASQUAL model. For caQTL mapping in L1 cell-types, we only included cell-types with at least 50 cells in at least 10 individuals. From the pseudobulk count table, we calculated library sizes and phenotype PCs (after scaling and centering; using 'prcomp' function in R). To get allelic-specific read counts, we extracted reads from each group using 'filterbarcodes' command from sinto v0.7.5 and counted allelic-specific reads using 'createASVCF.sh' from RASQUAL. We only kept bi-allelic SNPs with at least four minor allele counts across tested individuals. We included library size as offsets and five genotype PCs, the number of cells, and the GC content for each peak as covariates. RASQUAL was run in nominal mode and permutation mode. We extracted the lead SNP for each tested peak and used nominal and permuted $\log_{10}(q$ value) from RASQUAL to calculate empirical $p$ values with 'empPvals' function from the 'qvalue' R package, and then derived $q$ values from the empirical $p$ values. We used $q$ value below 0.1 as the cutoff for significant caQTLs.

#### Enrichment of RASQUAL caQTLs in bulk LCL caQTLs

To test the enrichment of RASQUAL caQTLs from WB in bulk caQTLs from LCL, we obtained summary statistics from a previous study.[28] We extracted lead SNP for each peak in the LCL dataset and ranked them by their significance. We then tested the enrichment of RASQUAL lead SNPs in the top 1%, 5%, and 10% of the most significant LCL caQTL lead SNPs using Fisher's exact test.

#### Enrichment of RASQUAL caQTLs in DICE bulk eQTLs

To test the enrichment of RASQUAL caQTLs from common immune-cell types in bulk eQTLs from DICE, we first extracted the genomic locations of lead eQTL SNPs and extended it by 500 bp on each side and converted it to bed format. These eQTL regions form the genomic annotation in which caQTL enrichment is tested. We then converted RASQUAL lead caQTL SNP positions to bed format, and tested their enrichment in eQTL regions using QTLtools 'fenrich' command,[34] while feeding our peak set into the '–tss' argument to adjust for the fact that RASQUAL caQTLs are enriched in the cPeaks themselves. The odd ratios from 'fenrich' were visualized after being normalized to the maximum odd ratio in each caQTL cell type.

#### Sharing of caQTLs by pi1 statistics and peak overlap

We calculated caQTL sharing defined as peak overlap in one or more cell types. We used a conservative estimation of caQTL sharing in that analysis. Specifically, we did not consider a caQTL as shared if its $q$ value is larger than 0.025 in another cell type, instead of the cutoff of 0.1 we used in single cell-type analysis. This choice of thresholds stems from multiple hypothesis testing adjustments. While $q$ value <0.1 defines significant caQTLs in a given cell type, assessing cell-type specificity requires further correction for the number of cell types analyzed. Since we focused on five major cell types (NK and other T cells had very few caQTLs), we applied a Bonferroni correction: 0.1 divided by (5-1) = 0.025. This means that for a caQTL with $q$ value <0.1 in cell type A to be considered shared, it must have $q$ value <0.025 in another cell type. This stringent approach yields a more conservative estimate of shared caQTLs compared to using qval <0.1 alone. Despite this conservative threshold, we still observed widespread sharing of caQTLs across immune-cell types (except monocytes), supporting our conclusion that caQTL sharing is pervasive.

To more directly quantify caQTL sharing with alternative methods, we calculated pi1 statistics[78] and peak overlap in all pairs of cell types. The pattern of pairwise caQTL sharing is largely consistent for peak overlap and pi1 statistics: CD4 T and CD8 T cells have more shared caQTLs; NK cells also have higher pi1 statistics with CD8 T cells. These results are consistent with the expectation that more similar cell types share more caQTLs. Pi1 statistics is slightly smaller than peak overlap, as it is more sensitive to power difference and potential lead SNP mismatch. But still, at least 30–40% of caQTLs are shared between a pair of cell types (Figure S10). Note that the pairwise analysis quantifies sharing between two specific cell types. In Figures 4A and 4B, we defined sharing as significant caQTL between any two or more cell types, which is by definition higher than pairwise sharing.

#### Single-cell Poisson mixed-effect model (sc-PME) caQTL mapping

Single-cell caQTL mapping with the sc-PME model was first performed in three studies separately. For continuous covariates, we included top five genotype PCs, top five LSI dimensions, TSS enrichment scores, fraction of mitochondrial reads, log10 of number

of unique fragments, all of which are scaled and fitted as fixed effects. We also included libraries and donors as random effects. The Poisson mixed effect model was fitted using the 'glmer' function ('family = poisson') in the lme4 R package. We set the additional options as 'nAGQ = 0, control = glmerControl(optimizer = "bobyqa," calc.derivs = F)' to save computational time for model fitting. We performed meta-analysis using effect sizes and standard errors from all SNPs in the three datasets and ran Metasoft without genomic control. For downstream analysis, we used effect sizes and standard errors from the random effects model and $p$ values from the Han and Eskin's Random Effects model (RE2).[33] To call significant caQTL in the meta-analyzed sc-PME results, we first applied Bonferroni correction for all SNPs in a given peak, extracted the lead SNP for each peak, and then calculated $q$ values from the Bonferroni-adjusted $p$ values across all lead SNPs. We used a $q$ value below 0.1 as a cutoff for significant lead caQTL.

### Dynamic caQTL mapping with sc-PME
To identify dynamic effects of lead caQTL SNPs along cell trajectories defined by topic modeling, we tested for interaction between genotype dosages and topic loadings. To avoid confounding dynamic caQTL with cell-type specific caQTL, we (1) mapped dynamic caQTLs separately in each common cell-type and (2) only included topics that are present in each cell-type as follows: B (k1, k11), CD4 T (k6, k7, k17), CD8 T (k3, k6, k7, k14, k17, k18, k19), NK (k3, k17), monocyte (k10, k12, k15, DC (k4, k10, k12, k15), other T cell (k3, k6, k8, k14, k17, k18, k19). For each SNP, we fitted a model with genotype-by-loading interaction term and a reduced model without the interaction term. We then used R function 'anova' to perform a likelihood ratio test (LRT) comparing the two models and used $p$ values from LRT to call significant dynamic caQTLs. Because we only tested the top caQTL SNP for every cPeak, we calculated $q$ values from LRT $p$ values in each topic separately, and then multiplexed the $q$ value by the number of topics in which a given SNP was tested for; this is equivalent to a Bonferroni adjustment on the number of topics tested for each SNP, thus caQTL SNPs from cells with more topics (e.g., CD8 T cells) were subjected to more stringent significant level cutoff. We reported adjusted $q$ value below 0.01 as significant dynamic caQTLs. Since we only have the interaction effects for lead caQTL SNPs, we annotated the colocalization states of dynamic caQTLs by examining the results from the linear sc-PME model.

### Permutation analysis for sc-PME model
One potential concern for sc-PME model is that it can lead to false positives, partially due to high sparsity in single-cell data. To validate that our sc-PME model is well-calibrated, we conducted permutation analysis. To do so, we analyzed both RASQUAL-defined cPeaks (which by definition contain significant caQTLs) and carefully matched non-cPeak controls with similar sparsity and mean counts. Our permutation tests showed excellent calibration: for cPeaks, 9.55% had $q$ values <0.1 under permutation. For matched non-cPeaks, 8.50% had $q$ values <0.1 under permutation (Figure S7). These results closely match the expected null distribution for both tested SNPs and lead SNPs, confirming our model's proper calibration.

### Comparison of sc-PME with QTLtools
In addition to permutation analysis, to demonstrate the validity of our sc-PME model in caQTL mapping, we compared its effect sizes with QTLtools,[34] which is the most widely used method for bulk QTL mapping. Focusing on the same set of peaks and mapping window size as our sc-PME model in monocytes, we found that sc-PME and QTLtools show high concordance (0.94) and correlation (0.78) in their effect sizes. Moreover, of the 16,017 significant cPeaks found in sc-PME, QTLtools only identified 9,253 of them, indicating that sc-PME is better powered than bulk linear model (Figure S8A and S8B).

### Sc-PME model captures distal caQTL effects on co-accessible peaks
We found that the majority of caQTLs identified by RASQUAL localize directly within or near chromatin accessibility peaks. For these proximal associations, the detected variants are highly likely to be the true causal SNPs regulating chromatin state. However, RASQUAL's narrow detection window inherently misses distal regulatory effects—a critical limitation, given that chromatin interactions often span distances greater than 50 kb. This is supported by well-established principles from *cis*-eQTL studies, where analyses routinely employ 1 Mb windows to account for distal regulation mediated by enhancer-promoter looping. Just as distal enhancers can physically interact with gene promoters to modulate expression, analogous mechanisms could generate distal caQTLs, where variants influence chromatin accessibility through long-range chromosomal contacts.

Recent work by Benaglio et al.[9] demonstrated that lead caQTLs often coordinately regulate both their target chromatin peak (cPeak) and nearby transcription start sites (TSS) co-accessible with that peak, providing a mechanistic basis for distal caQTL effects. We hypothesized that our sc-PME model could better detect these relationships because—unlike methods relying on haplotype phasing—it permits analysis across larger peak to TSS distances.

To test this, we analyzed monocyte caQTLs using a three-step approach: First, we identified significant cPeaks with caQTLs located within 1,000 bp of their peak centers. Next, for each of these "reference cPeaks," we used ArchR to detect co-accessible peaks (coAcc peaks) within 125 kb (correlation >0.5). Finally, we tested whether lead SNPs from reference cPeaks also influenced accessibility at these distal coAcc peaks.

The results confirmed our hypothesis: among 2,913 tested lead-SNP–coAcc-peak pairs, 204 showed significant genetic co-regulation (q < 0.1), with half (100/204) involving secondary cPeaks. Importantly, when we repeated the analysis using 3,000 distance-matched non-co-accessible control peaks (0.05 < correlation <0.1), only 6 showed evidence of co-regulation. Even among non-co-Acc peaks with nominal significance ($p < 0.05$), effect sizes and concordance with reference cPeaks were markedly weaker than for true coAcc peaks (Figure S9).

These findings confirm that caQTL effects can propagate at least 125 kb away from an accessibility peak, validating our sc-PME model's use of this window size while providing direct evidence for the long-range regulatory architecture.

## Colocalization of caQTLs with bulk eQTLs and GWAS

To perform colocalization between caQTLs and eQTLs, we used the eQTL summary statistics from the DICE study we re-processed and published before.[1] We tested for colocalization between a caQTL and an eQTL when there are more than 150 overlapping SNPs and their corresponding lead SNPs are among the overlapping SNPs.

To perform colocalization between our caQTL and in-sample COVID-19 eQTLs, we used the list of significant eGenes defined as mashr[32] local false sign rate (lfsr) below 0.1 in the accompanying manuscript (Randolph et al.). We tested for colocalization between a caQTL and an eQTL when there are more than 150 overlapping SNPs and their corresponding lead SNPs are among the overlapping SNPs.

To perform colocalization between caQTL and GWAS, we used GWAS summary statistics of 11 immune-related diseases and 36 blood cell-type GWAS that we accessed previously.[1] Briefly, we defined a GWAS locus as a 1 Mb window centered around an SNP with a $p$ value below 1e−7, starting from the SNP with the smallest $p$ value, removing all SNP lies within 1 Mb, and iteratively identified all GWAS loci until no SNPs with $p$ value below 1e−7 remained. Like eQTL, we tested for colocalization between a caQTL and a GWAS locus when their corresponding lead SNPs are among the overlapping SNPs and there are more than 150 overlapping SNPs. All colocalization analyses were performed with the 'coloc.abf' function from R package 'coloc' v5.2.1.[29]

### Characterization of GWAS-QTL colocalizations

Several factors can potentially explain the lack of colocalization between caQTLs and eQTLs, including the effect sizes of QTLs, whether they have similar enrichment in certain genomic annotations (TSS, intergenic regions etc.).

We found that, as expected, caQTLs colocalized with eQTLs are closer to TSS compared to those that do not colocalize (Figure S12A). This might be because that eQTLs tend to lie closer to TSS. However, this difference is small, and we did not observe enrichment of colocalized cPeaks in promoter peaks compared to uncolocalized cPeaks.

We compared the Z-scores of caQTLs that colocalized with eQTLs, did not colocalize with eQTL and those that are not tested for colocalization. Colocalized caQTLs tend to have the largest $Z$ scores, followed by uncolocalized caQTLs. This difference is larger in B cells, which has the smallest number of cells, therefore lower statistical power (Figure S12B). This observation is consistent with our previous eQTL-GWAS analysis.[1] However, many uncolocalized caQTLs showed $Z$ scores that are comparable to colocalized caQTLs. This means that power only partially explains the lack of colocalization between caQTL and eQTL.

Finally, as eQTL tend to be closer to TSS, whereas caQTL can capture distal regulatory elements, we compared if colocalized cPeaks are systemically different from non-colocalized cPeaks in terms of genomic annotation (i.e., promoter, exonic, intronic, and intergenic). There is no significant difference between colocalized and non-colocalized cPeaks (Figure S12C).

