## [Document S2. Transparent peer review records for Mu et al. · Cell Genomics]

Impact of disease-associated chromatin accessibility QTLs across immune cell types and contexts

Author list

Zepeng Mu (牟泽鹏), Haley E. Randolph, Raúl Aguirre-Gamboa, Ellen Ketter, Anne Dumaine, Veronica Locher, Cary Brandolino, Xuanyao Liu, Daniel E. Kaufmann, Luis B. Barreiro, Yang I. Li

Summary

Initial submission: Received : December 27th 2024

Scientific editor: Judith Nicholson

First round of review: Number of reviewers: 3
Revision invited : April 7th 2025
Revision received : July 22nd 2025

Second round of review: Number of reviewers: 3
Accepted : 10th October 2025

Data freely available: Yes

Code freely available: Yes

This transparent peer review record is not systematically proofread, type-set, or edited. Special characters, formatting, and equations may fail to render properly. Standard procedural text within the editor's letters has been deleted for the sake of brevity, but all official correspondence specific to the manuscript has been preserved.

Referees' reports, first round of review

Reviewer 1:

The stated primary goal of this paper was to map chromatin accessibility QTL (caQTL) in blood cell types determined by single-cell ATAC-seq in 48 individuals with 59 samples. Co-localization with GWAS loci determined significantly more links than found in eQTL analyses. The authors argue these caQTL provide missing links to disease that are missed by eQTL. While there are some potentially significant findings, there are several major concerns that reduce enthusiasm for this manuscript.

Major concerns:

1) Though it is clearly and well-stated in the Abstract and even the Introduction that the main purpose of this manuscript is a caQTL analysis to understand GWAS variants, the paper has many very unrelated results sections and discussions that detract from this focus and result in a fragmented and often confusing paper.

For instance, even though not mentioned in the Abstract or Introduction, the samples were primarily taken from COVID patients with various phenotypes. These phenotypes were then used in several of the analyses. But this is a completely different direction than the stated main goal. The authors should either remove these COVID analyses or more fully integrate the COVID goal into the paper.

2) A bit unconventionally, targeted genotype experiments were not performed on the majority of the patients, but rather the genotypes were inferred from the ATAC-seq data using GLIMPSE. The quality of these imputed genotypes is not clear. In Fig 1H, the legend states these are compared to low-pass WGS, but this data is not described and not presented in this figure. Imputation quality scores are presented, but this does not represent a true measure of accuracy, nor are the dosages or imputed SNPs (Ext Fig 1g). Since there are direct measures of

variants from a small set of samples using a genotyping microarray, then it should be determined what the sensitivity and specificity of the ATAC-seq inferred genotypes on these variants in these samples. If, as Ext Fig 1F suggests, that relatively few SNPs are covered by at least 3 reads, this calls into question the accuracy of the genotypes, and thus all of the QTL analyses.

3) caQTLs are called using RASQUAL, and then some correlative analysis is performed that supports this analysis, but it certainly does not provide evidence of a false positive rate. But then since, as the authors state, "RASQUAL does not report effect size and standard error", they then use a different model (sc-PME) to generate summary statistics. And they state a larger window (250Kb) is used. The validity of this model is based on replicability of RASQUAL results, which at 79% is good but not great, but then the fact that more distal sc-PME caQTLs are enriched in cPeaks (where genotypes are based on data on ATAC-seq peaks) does not provide solid evidence. And what is the biological interpretation of a SNP affecting an open chromatin region that is >50Kb away?

4) The increased sharing of caQTLs by the scPME analysis compared to RASQUAL and previously reported analyses is explained away by a discrepancy in statistical power. It isn't clear why the scPME method, which was not independently validated, has more power than RASQUAL. It more efficiently can test more SNPs, but this does not equate to increased power. It's also unclear how the authors define "sharing" of their sc-PME caQTL. SNP overlap? Colocalization? Another method? This is not stated in the Methods.

5) Statements by the authors that caQTL that are not eQTL are "difficult to interpret" suggest the authors may not fully understand the relationship between accessible chromatin and gene expression measured at a snapshot in time, especially poised accessible chromatin whose activity may only manifest under certain conditions not present at this snapshot.

Minor:

1) It isn't clear why results for multiple total topics (k=6, 10, 20) are presented if

k=20 is primarily used for most analyses. The others don't seem relevant. And if topic k2 seems to be related to data quality, why isn't this technical variation removed?

2) It states that 4 different peak-to-gene mapping strategies were tested, but the results in Ext Fig 3C are not explained and it isn't clear what the criteria was to determine the "best" nor how the different methods worked.

3) scATAC-seq analysis define both "L1" and "L2" annotations, but it appears that only L1 annotations are meaningfully used for caQTL and other analyses. What is the purpose of the L2 annotations given the stated goal of this paper?

4) Line 383: The authors state "This suggests that the majority of caQTLs do not influence the expression level of a nearby gene in the many immune cell-types and contexts included in the DICE dataset." This statement seems misleading. Colocalization indicates if the genetic signals are likely shared. There's no causal inference. If most of the caQTL peaks are found to overlap enhancer regions, and the DICE eQTL are likely underpowered, could this mean that DICE is mostly detecting nearby, promoter-enriched eQTL, and so could be another possible explanation as to why there are not many overlaps?

5) Line 375: If the colocalization results between the caQTL and COVID eQTL were largely underwhelming, is it worth reporting/including?

6) Line 393: MS4A4A monocyte eQTL: did you do any analysis to colocalize among DICE eQTL to determine eQTL sharing?

7) Line 429: why not look for ABC linked genes?

8) Line 460: include coloc PPs for caQTL/eQTL

9) Line 467: include coloc PPs for caQTL

10) Line 478: "we focused on cPeaks overlapping with TSS": Why was this criteria imposed?

11) Line 481: For consistency, include coloc PPs

12) Line 495: How was dynamic caQTL colocalization performed?

13) Line 524: Conclusion not well supported. caQTL (COVID patients) and eQTL (DICE - presumably healthy) were not identified in the same context, so other possible explanation besides pleiotropy.

14) Figure 1g: would be good to include study information in addition to disease

state so that can see if study was related to groupings

15) Figure 2b: what is the dashed box?

16) Figure 2d: Typo? Is it supposed to be k1 instead of k2?

17) Figure 4a: the q-value thresholds used to determine cell type specific caQTL are confusing. $q < 0.1$ is considered a significant caQTL in a given cell type, but if the q-value is > 0.025 in other cell types, the caQTL is considered cell type specific even though it would technically considered significant in the other cell type.

18) Figure 4b: top plot, why use log10 scale?

19) Figure 4 legend: d and e flipped

20) Figure 5g: Where is the NK caQTL locus plot? Mentioned to colocalize in main text (line 549).

21) Figure 5 legend: missing description for plot e.

22) Extended Figure 1d: include study information in addition to disease state?

23) Extended Figure 5c: Are these eQTL or caQTL signals?

24) Extended Figure 5e: Include plot for CD8? Mentioned in text (line 487)

25) Extended Figure 6: An opportunity for an UpSet plot to better visualize sharing?

26) Some general comments about the all of the LocusZoom plots in main and extended figures:

a. x-axes should be aligned for all stacked plots for easier interpretation

b. Similarly, in "shared" cases (i.e. shared caQTLs/eQTLs), y-axes should be standardized

c. For stacked plots, a single index variant should be selected and highlighted in all plots so that LD variants are colored consistently for easier interpretation.

d. Include an LD color legend

e. Gene tracks should be added to include the target gene or eGene and also give a reference for where cPeaks are relative to the gene.

f. When peak tracks are included, add y-axis

g. For all LocusZoom plots include cell type and analysis (i.e. caQTL/eQTL)

Reviewer 2:

This manuscript provides a comprehensive study on chromatin accessibility QTLs (caQTLs) across immune cell types, utilizing single-cell ATAC-seq data. The

authors constructed a harmonized chromatin accessibility map from 282,424 cells across multiple conditions, identified extensive caQTLs and analysed their relations with eQTLs and GWAS loci. The study sheds light on the functional effects of caQTLs and their relevance to interpret disease associated genetic variants, emphasizing the need to expand eQTL studies in disease-relevant contexts. The computational approaches are rigorously designed and the results look solid, leading to some novel insights into the quantitative understandings of the genetic basis of gene regulation under specific cellular contexts. I just have a few questions for the authors to further clarify.

Major Comment:

1. The inferred topics from topic models are not necessarily cell types or subtypes. The topics may correspond to variations of co-factors and batch effects. The authors need to provide a justification to clear this potential issue.
2. About the colocalization analysis of caQTLs and eQTLs, could the authors show the distribution of their effect sizes respectively and check whether there is a bias caused by smaller effect sizes of subsets of non-overlapping QTLs?
3. For caQTLs that do not colocalize with eQTLs, how is the genomic distance distribution between these caQTLs to their nearest gene promoters?
4. For cell-type specific caQTL-eQTL colocalizations (versus shared caQTL-eQTL), in addition to the enrichment with lineage-associated TF motifs, I wonder whether the authors could check whether specific TF motif combinations are enriched? The combinatorial TF patterns may represent the regulatory grammar for precise cell-type specific regulation.
5. On Page 10, lines 399-417: it is confusing to interpret the importance of caQTL-eQTL pair colocalize in multiple cell types as the strongest feature to predict cell-type specific caQTL-eQTL colocalizations. If it is cell-type specific, why does the multi-cell-type sharing feature become the strongest feature?

Minor Comment:

1. Some annotations in the manuscript remain unclear, such as the terms "(MONO%, MONO#, GRAN%MYELOID)" on page 5, which need further clarification or definition. Additionally, there are still some typographical and grammatical errors throughout the text that should be addressed to improve readability and professionalism.
2. The choice of colors in some figures, such as Figure 3d and Figure 3j, makes it difficult to discern finer details. Enhancing the contrast and clarity of these figures would significantly improve their interpretability and overall quality.

Reviewer 3:

Overview: Mu et al identify and analyze chromatin accessibility (ca)QTLs using harmonized datasets of single-cell ATAC data from 59 samples (48 unique donors). This work has two main contributions. Firstly, they provide a pipeline to harmonize cell type annotations across multiple data sources, followed by topic analysis to demonstrate patterns of chromatin accessibility across cell types and cell states. Second, the authors perform caQTL mapping and systematically evaluate colocalization with eQTLs and GWAS loci for relevant traits. Their approach and results are highly likely to be of interest to the statistical genetics/genomics communities and provide insights on interpreting GWAS risk loci using multi-omics approach. This work has important implication on the lack of cellular context in eQTL mapping and more robust causal inference for GWAS risk loci when observing convergence of both modalities. The manuscript is written well and results presented clearly. With that said I have several comments.

Major Comments:

1. Regarding the COVID-19 example, the author identified two groups of COVID-19 patients have more NK or monocyte cells as presented in Figure 1g. However later when using topic analysis, the authors concluded the k17 topic represent CD8 TEM and its loading tracks with COVID-19 cells proportions, thus suggesting an expanded CD8 TEM population in COVID-19 patients. The authors should comment on why NK or monocyte expansion is absent in the topic analysis and why this discrepancy can occur.
2. I suggest the authors first describe ancestry, age, and sex for in-sample individuals to provide the generalizability context of results produced by this work, especially for colocalization results with external dataset (e.g., COVID19-

patients in-sample vs. all healthy individuals in DICE dataset). For example, the PCA plot shows the 20 individuals collected by this study are either European- or African-ancestry. Based on accompanied paper Randolph et al, these 20 individuals are all males.

3. For the caQTL mapping, the authors found extensive sharing of caQTLs using Poisson mixed effect in contrast to more cell-type specific results using RASQUAL. Since the sample size is relatively small and peak counts are sparse in this analysis, the authors should demonstrate that PME model is well-calibrated under null by permuting peak counts and show distribution of p values. The authors should also explain why age, sex and COVID-19 disease status are not considered in the caQTL model.

4. The authors used logistic model to predict colocalization between caQTL and eQTLs using multiple predictors, but not including TSS distance. Can the authors plot and describe the distance of lead caQTL to TSS of colocalized gene (i.e., eQTLs) and for those not colocalized respectively? Assuming promoter effect of eQTL is likely shared between cell types, I would expect the majority of colocalized ones are close to TSS. The non-colocalized ones are likely distal where eQTL signals are lacking due to limited eQTL sample size or lack of relevant context.

5. The in-sample COVID-19 eQTL mapping from Randolph et al. was performed for controlled samples and IAV-infected samples respectively. Can the authors clarified on which results are used in this work? If using both, the authors should consider present their colocalization results between caQTLs and these condition-specific eQTLs separately.

Minor Comments:

1. For Figure 1g, it's not clear which two groups of individuals have expanded NK or monocyte population. Can the authors highlight those groups as done in Extended Figure 1d?
2. For Figure 3i, consider order the cell type on x-axis by their total number of cells to show their relationship with number of identified dynamic caQTLs.
3. For the top bar plot in Figure 4c, the y axis name should be "Number of

colocalized caQTLs".

4. The authors should clarify on the interpretation on Figure 4d and 4e as it shows mixed message as written in line 407-409. The cell-type specific colocalization in Figure 4d is likely driven by difference in lymphocytes (eg. T cells) versus monocytes. The strong predictor of "COLOC in another cell type" in Figure 4e is likely reflecting similar subtypes as CD4 T cells and CD8 T cells.

5. For caQTL mapping in the 25 individuals with repeated measures collected by this study, can the authors clarify on how different time point are modeled? Or did the authors use one time point?

6. Figure 5e lacks description in the caption.

Authors' response to the first round of review

Response to Reviewers' Comments

Reviewer #1: The stated primary goal of this paper was to map chromatin accessibility QTL (caQTL) in blood cell types determined by single-cell ATAC-seq in 48 individuals with 59 samples. Co-localization with GWAS loci determined significantly more links than found in eQTL analyses. The authors argue these caQTL provide missing links to disease that are missed by eQTL. While there are some potentially significant findings, there are several major concerns that reduce enthusiasm for this manuscript.

Thank you for the positive comments and concerns.

Major concerns:

1) Though it is clearly and well-stated in the Abstract and even the Introduction that the main purpose of this manuscript is a caQTL analysis to understand GWAS variants, the paper has many very unrelated results sections and discussions that detract from this focus and result in a fragmented and often confusing paper.

For instance, even though not mentioned in the Abstract or Introduction, the samples were primarily taken from COVID patients with various phenotypes. These phenotypes were then used in several of the analyses. But this is a completely different direction than the stated main goal. The authors should either remove these COVID analyses or more fully integrate the COVID goal into the paper.

We appreciate your comment. While the main purpose of our study is to perform a caQTL analysis to better understand the regulatory mechanisms underlying GWAS variants, we believe that the inclusion of COVID-19-related analyses enhances the biological relevance of our findings rather than detracting from the main focus.

Specifically, in **Figure 2**, we identified a topic in CD8 T cells that is significantly associated with COVID-19 status. Importantly, we demonstrate that this topic is enriched for dynamic caQTLs, directly linking genetic regulation to disease phenotypes. We believe that this signal would likely be absent or considerably weaker if our study were restricted to healthy individuals, thereby missing a crucial aspect of context-dependent regulation.

That said, we acknowledge that the COVID-19 context and related phenotypic analyses were not sufficiently integrated into the manuscript. In response to this feedback, we have revised our main text to better incorporate COVID-19-related analysis with caQTL analysis and situate the reader. Specifically we wrote the following on **lines 77**:

Though our study is not focused on COVID19, our data include 20 COVID-19 patients and 8 convalescent donors.

And lines 247:

While our study was not designed to investigate COVID-19, the significant heritability enrichment in accessible peaks linked to topic k17 led us to explore potential associations with COVID-19 status.

And lines 291:

Based on our observations that COVID-19 and healthy samples exhibited minimal differences in both cell clustering patterns and topic model distributions (with only k17 showing significant association), we performed caQTL mapping using a combined dataset without separating cells by disease status.

2) A bit unconventionally, targeted genotype experiments were not performed on the majority of the patients, but rather the genotypes were inferred from the ATAC-seq data using GLIMPSE. The quality of these imputed genotypes is not clear. In Fig 1H, the legend states these are compared to low-pass WGS, but this data is not described and not presented in this figure. Imputation quality scores are presented, but this does not represent a true measure of accuracy, nor are the dosages or imputed SNPs (Ext Fig 1g). Since there are direct measures of variants from a small set of samples using a genotyping microarray, then it should be determined what the sensitivity and specificity of the ATAC-seq inferred genotypes on these variants in these samples. If, as Ext Fig 1F suggests, that relatively few SNPs are covered by at least 3 reads, this calls into question the accuracy of the genotypes, and thus all of the QTL analyses.

Thank you for highlighting these comments and concerns. In **Figure 1h**, we mistakenly stated that GLIMPSE imputation was compared with low-pass WGS. This was inadvertently referring to a set of samples with low-pass WGS data that were no longer included in the dataset. Instead, Fig 1H was comparing genotyping from scATAC-seq reads with those from microarray (Benaglio) as ground truth. We apologize for our oversight.

The imputation quality score shown in **Figure 1h** is commonly used to benchmark new imputation methods, see Rubinacci et al., Nature Genetics, 2021 and Li, Mazur et al., Genome Research, 2021. Although these studies did not assess the performance of imputed genotype in QTL analysis, low imputation quality will under-estimate the effect size in QTL analysis. Thus, we only included SNPs with a quality score > 0.7 in our QTL analysis, which is commonly used in QTL analysis and does not have a huge impact on QTL analysis.

To fully address your concerns, we performed the following analysis:

Similar to previous **Figure S1g**, we calculated correlation between SNPs imputed from scATAC-seq reads and genotyped (not imputed) in the microarray data (Benaglio et al). We also repeated this analysis in samples we collected in this study. This is now the updated **Figure S1g**. We found that SNP dosages imputed from scATAC reads have higher correlation with ground truth genotypes from microarray compared to those imputed from microarray. Thus, we conclude that our SNP calls are as good or better than microarray-imputed SNPs, allowing us to use these for QTL mapping. Furthermore, in our caQTL mapping, we only used SNPs whose

minor allele frequencies are larger than 0.01 in the 1000G reference panel. Using this cutoff, the r^2 between our scATAC-imputed and microarray-imputed genotype is above 0.95.

Similar to Rubinacci et al, we calculated discordance rate at heterozygous SNPs. Overall, less than 5% of all SNPs are discordant with microarray data – for both genotyped and imputed SNPs. These results demonstrate that we can accurately impute genotypes from scATAC data. This is now updated as **Figure S1h**.

Although we observed that some SNPs were covered by three reads in a single-cell, in **Figure S1f**, these account for only a very small proportion of SNPs, and their impact on imputation are even smaller when reads from all cells from an individual are pooled together. Therefore, the contribution of an extra (erroneous) read from any given cell to the genotype is negligible.

Finally, we note that a recent preprint (Wenz et al., *Genome Biology*, 2025) also performed genotyping using bulk ATAC-seq reads in 653 public datasets for caQTL mapping, further supporting the validity of our methodology. To get a sense of genome-wide coverages of scATAC-seq versus bulk ATAC-seq data, we analyzed bulk ATAC-seq libraries from peripheral immune cells published in Calderon et al., *Nature Genetics*, 2019. We selected the bulk libraries with the largest and smallest number of reads from that study and compared coverage with our scATAC-seq.

To our surprise, aggregated scATAC libraries have better coverage than bulk ATAC-seq libraries. This may be due to the optimization of bulk ATAC-seq protocols to use a small number of cells followed by shallow sequencing. Altogether, these analyses indicate that our genotyping from scATAC should be at least as good as those from bulk ATAC-seq, including Wenz et al.

We have added a new section describing these analyses in the main text (**lines 139**):

Even though one of the studies (You et al.) did not genotype individual donors, we reasoned that we could approximate low-pass whole-genome sequencing (WGS) and/or bulk ATAC coverage by aggregating single-cell reads from each individual and adapt GLIMPSE15 to accurately impute common variants in these individuals (Figure S1e). We tested this by comparing microarray genotyped SNPs to the imputed SNPs in scATAC-seq using our GLIMPSE workflow (Methods). We found that genotype dosages imputed from scATAC reads and genotyping arrays were highly correlated across all reference minor-allele frequency (MAF) bins (>91%, Figure 1h, Figure S1f,g), indicating that imputed genotypes from scATAC-reads are highly accurate and are not biased by allelic imbalance in chromatin accessibility. We also assessed genotype imputation accuracy and observed a low discordance rate of approximately 5% across SNPs. This error rate was consistent with that observed when using microarray-based imputed genotypes. (Figure S1h). This allowed us to merge genotype likelihood estimates from all three studies and to perform joint imputation using GLIMPSE, resulting in a harmonized callset of 6.75 million high-quality SNPs for caQTL mapping (Figure S1i).

3) caQTLs are called using RASQUAL, and then some correlative analysis is performed that supports this analysis, but it certainly does not provide evidence of a false positive rate. But then since, as the authors state, "RASQUAL does not report effect size and standard error", they then use a different model (sc-PME) to generate summary statistics. And they state a larger window (250Kb) is used. The validity of this model is based on replicability of RASQUAL results, which at 79% is good but not great, but then the fact that more distal sc-PME caQTLs are enriched in cPeaks (where genotypes are based on data on ATAC-seq peaks) does not provide solid evidence. And what is the biological interpretation of a SNP affecting an open chromatin region that is >50Kb away?

Thank you for this comment. In terms of the comparison of RASQUAL and sc-PME, we would like to argue that 79% replicability across different QTL mapping methods is comparable to that observed in other studies (GTEx consortium, Nature, 2017, Extended Data Figure 4c). To further verify the validity of the sc-PME model, we mapped pseudobulk caQTL using FastQTL (QTLtools), the most widely-used and standard pipeline for QTL analysis, and compared the results with both RASQUAL and sc-PME results.

We used QTLtools to map caQTL in monocytes with the same window size and on the same set of RASQUAL cPeaks as tested in sc-PME, and identified 10,596 cPeaks, of which 9,253 were also found in sc-PME. sc-PME identified an additional 6,764 cPeaks, suggesting that sc-PME has higher power even when only RASQUAL cPeaks are tested. The effect size between QTLtools and sc-PME is also highly correlated and concordant.

While RASQUAL only analyzes variants within a limited genomic window, it can still detect distal regulatory effects. This occurs because the identified signal may actually reflect long-range regulatory interactions, where SNPs within the analyzed window are in weak linkage disequilibrium with the true distal causal variants. This mirrors how allele-specific eQTLs are mapped in RNA-seq: while only exonic reads are directly measured, the detected signal often reflects regulation by distal elements (e.g., promoters or enhancers) that genetically control gene expression (Mohammadi et al., Genome Research 2017).

In our study, we found that the majority of caQTLs identified by RASQUAL localize directly within or near chromatin accessibility peaks. For these proximal associations, the detected variants are highly likely to be the true causal SNPs regulating chromatin state. However, RASQUAL's narrow detection window inherently misses distal regulatory effects—a critical limitation, given that chromatin interactions often span distances greater than 50 kb. This is supported by well-established principles from cis-eQTL studies, where analyses routinely employ 1 Mb windows to account for distal regulation mediated by enhancer-promoter looping. Just as distal enhancers can physically interact with gene promoters to modulate expression, analogous mechanisms could generate distal caQTLs, where variants influence chromatin accessibility through long-range chromosomal contacts.

Recent work by Benaglio et al. (PLoS Genetics 2023) demonstrated that lead caQTLs often coordinately regulate both their target chromatin peak (cPeak) and nearby transcription start sites (TSS) co-accessible with that peak, providing a mechanistic basis for distal caQTL effects. We hypothesized that our sc-PME model could better detect these relationships because—unlike methods relying on haplotype phasing—it permits analysis across larger peak to TSS distances.

To test this, we analyzed monocyte caQTLs using a three-step approach: First, we identified significant cPeaks with caQTLs located within 1,000 bp of their peak centers. Next, for each of these "reference cPeaks," we used ArchR to detect co-accessible peaks (coAcc peaks) within 125 kb (correlation > 0.5). Finally, we tested whether lead SNPs from reference cPeaks also influenced accessibility at these distal coAcc peaks.

The results confirmed our hypothesis: among 2,913 tested lead-SNP-coAcc-peak pairs, 204 showed significant genetic co-regulation ($q < 0.1$), with half (100/204) involving secondary cPeaks. Importantly, when we repeated the analysis using 3,000 distance-matched non-coaccessible control peaks ($0.05 < \text{correlation} < 0.1$), only 6 showed evidence of co-regulation. Even among non-coAcc peaks with nominal significance ($p < 0.05$), effect sizes and concordance with reference cPeaks were markedly weaker than for true coAcc peaks.

These findings confirm that caQTL effects can propagate at least 125 kb away from an accessibility peak, validating our sc-PME model's use of this window size while providing direct evidence for the long-range regulatory architecture proposed by Benaglio et al.

All these new analyses are now described in Supplementary Notes and referenced in our revised manuscript (lines: 360).

More broadly, caQTL SNPs tend to exhibit consistent effects on distal peaks that are co-accessible, offering mechanistic insight into how distant caQTLs identified using the sc-PME model may function (Supplementary Notes).

4) The increased sharing of caQTLs by the scPME analysis compared to RASQUAL and previously reported analyses is explained away by a discrepancy in statistical power. It isn't clear why the scPME method, which was not independently validated, has more power than RASQUAL. It more efficiently can test more SNPs, but this does not equate to increased power. It's also unclear how the authors define "sharing" of their sc-PME caQTL. SNP overlap? Colocalization? Another method? This is not stated in the Methods.

We appreciate this insightful comment. The increased statistical power of single-cell methods is well-supported by recent literature. As demonstrated in Zhou et al. (medRxiv, 2024), single-cell Poisson models consistently outperform pseudobulk approaches in eQTL detection, achieving greater power through two key advantages: (1) more efficient use of all available cellular measurements and (2) explicit modeling of intra-sample cellular heterogeneity. Multiple other studies (Cuomo et al., Genome Biology, 2021, Valencia et al., bioRxiv, 2024) have similarly shown that single-cell methods can detect signals missed by bulk approaches.

While we didn't emphasize this comparison in our original manuscript (as these findings represent established work in the field), we've now added discussion of this important methodological consideration on lines 642:

The sc-PME approach shows strong concordance with pseudobulk methods while outperforming tested alternatives – including RASQUAL, QTLtools, and sc-lme – due to its use of all available cellular measurements and explicit modeling of intra-sample heterogeneity.

In terms of caQTL sharing, in **Figure 4 a,b** we defined it as peak overlap. We agree that peak overlap does not necessarily mean SNP overlap. To better quantify caQTL sharing, we calculated peak overlap and Pi1 statistics in all pairs of cell types, and compared the two results. The pattern of pairwise caQTL sharing is largely consistent for peak overlap and Pi1 statistics. CD4 T and CD8 T cells have more shared caQTLs. NK cells also have higher Pi1 statistics with CD8 T cells. These results are consistent with the expectation that more similar cell types share more caQTLs. Pi1 statistics is slightly smaller than peak overlap, as it is more sensitive to power difference and potential lead SNP mismatch. But still, at least 30-40% of caQTLs are shared between a *pair* of cell types. We note that the pairwise analysis quantifies sharing between two specific cell types. In **Figure 4a, b**, we defined sharing as significant caQTL between **any** two or more cell types, which is by definition higher than pairwise sharing.

Overall, we reached the same conclusion that caQTLs are extensively shared using multiple ways to quantify sharing.

We added these results to Supplementary Notes.

5) Statements by the authors that caQTL that are not eQTL are "difficult to interpret" suggest the authors may not fully understand the relationship between accessible chromatin and gene expression measured at a snapshot in time, especially poised accessible chromatin whose activity may only manifest under certain conditions not present at this snapshot.

We appreciate this thoughtful comment and completely agree that accessible chromatin regions—including poised enhancers—can play regulatory roles that may not be immediately evident in steady-state gene expression measurements. When we described caQTLs without corresponding eQTL effects as "difficult to interpret," we did not intend to suggest they are biologically irrelevant. Rather, our point was that interpreting GWAS loci—even after identifying a colocating caQTL—can remain challenging in the absence of clear eQTL support in the same cellular context.

This aligns with one of our manuscript's conclusions: while caQTLs provide valuable insights into regulatory variation, those without detectable eQTL effects warrant careful interpretation, particularly in GWAS colocalization analyses. This caution reflects not a lack of mechanistic understanding, but rather the inherent complexity of chromatin accessibility's relationship to gene regulation, which is often context-dependent and dynamic. We revised the text to clarify this perspective and better emphasize the biological significance of these regulatory relationships. Lines 527:

While these variants clearly influence chromatin accessibility, their lack of effect on gene expression in the studied cell types makes it difficult to identify their target genes or relevant cellular contexts. These loci may represent poised regulatory elements that require additional cellular stimuli or developmental stages to exert their effects on transcription.

Minor:

1) It isn't clear why results for multiple total topics ($k=6, 10, 20$) are presented if $k=20$ is primarily used for most analyses. The others don't seem relevant. And if topic k_2 seems to be related to data quality, why isn't this technical variation removed?

Thanks for this comment. The number of topics to fit is a hyperparameter that has to be decided somewhat arbitrarily before fitting a topic model. In the main text, we briefly touched on how the loadings change depending on the number of topics and explained our rationale for choosing $k=20$ in all downstream analysis. Importantly, because the topic modelling problem is non-convex, each run will give rise to topics in different orders, making it difficult to compare models with different K . To overcome this, we started from a smaller number of topics and progressively added more topics to be fitted while conditioning on already-fitted topics. **Figure 2a** is explicitly showing this progress (please also see Methods).

In terms of topics k_2 , we believe it important to show it even if it represents data quality rather than ignoring it. Because we did not perform any batch correction before fitting the topic model, we expect some topics to reflect data quality and batch effects. In fact, one desirable feature of the topic model is that it can separate technical and biological variations into distinct topics, making it straightforward to only use biologically relevant topics in downstream analysis. We have added sentences in **Supplementary Notes** to emphasize this feature, including for topics k_2 , k_5 , k_9 , k_{16} .

Lines 191:

We identified several additional topics (k_5 , k_9 , k_{16} , k_{19}) that most likely reflect technical variation, including data quality and batch effects (**Figure 2b**, **Supplementary Notes**). This observation supports the model's ability to effectively partition biological signals from technical artifacts into distinct topics. Importantly, this separation enables selective focus on biologically meaningful topics for subsequent analyses while minimizing confounding technical influences.

2) It states that 4 different peak-to-gene mapping strategies were tested, but the results in Ext Fig 3C are not explained and it isn't clear what the criteria was to determine the "best" nor how the different methods worked.

To provide more details on these four strategies, we added a section to Supplementary Note detailing how we implemented them. We also performed new analyses to better compare the four strategies:

Our investigation of well-known naive T cell marker genes revealed three key findings across the different peak-to-gene mapping approaches: First, these markers consistently showed higher scores in the naive T cell topic (k7) compared to other topics for all approaches. Second, established marker genes such as *CCR7* and *LEF1* consistently outperformed less specific genes like *CD8A* within each strategy. However, our third and most interesting observation revealed that while these markers performed well in absolute terms, they ranked surprisingly lower in the TSS and nearest TSS peak methods due to numerous other genes achieving even higher scores.

Motivated by these observations, we performed gene-set over-representation analysis on top 1000 genes with the highest gene score for each peak-to-gene mapping method.

Our analysis of the top five enriched gene sets for each method revealed important differences in biological relevance. The exponential decay scoring method showed significant enrichment for key T cell pathways including "T cell differentiation" and "T cell activation," while methods relying on TSS proximity or nearest peak association were predominantly enriched for mRNA processing pathways. This pattern held when examining MSigDB immune hallmark gene sets (C7 collection), where our scoring method and the co-accessibility approach consistently identified gene sets with T cell upregulation functions, confirming their T-cell-specific expression patterns. In contrast, the TSS-based methods produced enrichment of biologically irrelevant gene sets for T cell biology. Together, these results demonstrate that our exponential decay gene scoring method provides substantially more biologically meaningful results than simply linking peaks based on TSS proximity alone.

We added these results to Supplementary Notes.

3) scATAC-seq analysis define both "L1" and "L2" annotations, but it appears that only L1 annotations are meaningfully used for caQTL and other analyses. What is the purpose of the L2 annotations given the stated goal of this paper?

Thank you for the comment. We did not use L2 annotation in caQTL mapping as the smaller number of cells reduces statistical power for QTL mapping. The L2 annotations were primarily used to evaluate the resolution and quality of our scATAC-seq data, particularly for identifying relatively rare cell subtypes. As shown in **Figure 1f**, we observed distinct chromatin accessibility patterns around gene bodies of well-established marker genes at the L2 level, supporting the biological relevance of these finer annotations.

However, due to the smaller number of cells within many of the L2 cell types, we chose to perform caQTL mapping using the more common L1 annotations to ensure sufficient statistical power. Thus, while L2 annotations were not directly used in downstream QTL analyses, they served an important role in validating the granularity and robustness of our single-cell data, and was used in later analysis such as **Figure 4g**.

We have updated the main text to clarify this rationale in the manuscript to better explain the role and value of the L2 annotations in the context of our study's primary goals (lines 131):

We note that while L2 annotation provides enhanced cellular state resolution, its utility for quantitative analysis is limited because of reduced statistical power due to smaller cell counts per state. Consequently, we employed L1 annotations for our primary QTL mapping to maintain statistical power, reserving L2 annotations for specific analyses where finer cellular resolution was required.

4) Line 383: The authors state "This suggests that the majority of caQTLs do not influence the expression level of a nearby gene in the many immune cell-types and contexts included in the DICE dataset." This statement seems misleading. Colocalization indicates if the genetic signals are likely shared. There's no causal inference. If most of the caQTL peaks are found to overlap enhancer regions, and the DICE eQTL are likely underpowered, could this mean that DICE is mostly detecting nearby, promoter-enriched eQTL, and so could be another possible explanation as to why there are not many overlaps?

Thank you for this comment. We agree with your intuition that DICE is mostly detecting nearby, promoter-enriched eQTL. In fact, we have shown exactly this in our previous work comparing DICE with a much larger eQTL data, DGN (Mu et al., *Genome Biology*, 2021). To address the reviewer's concern about power, we performed two new analyses.

In the first analysis, we used eQTLs identified from the eQTLGen consortium ($n = 31,684$ individuals), which should be well powered to detect eQTL at enhancers. Across all five cell types, 9,346 (32.4%) of all caQTLs colocalized with eQTLGen data, nearly doubling the number of colocalized caQTL compared to DICE (4,088). This is expected given the much larger sample size of eQTLGen. Still, ~70% of caQTLs are not detected as eQTL. Therefore, our conclusion that "the majority of caQTLs do not influence the expression level of a nearby gene" remains.

We added the following sentences to the main text (lines 430):

The most colocalization signals found was with eQTLGen whole blood eQTLs (9,346 caQTLs, 32.4%), attributable to the enhanced statistical power of this large dataset. Nevertheless, we prioritized DICE-based colocalizations for downstream analyses as they provide cell-type-specific resolution.

In the second analysis, since DICE might be biased toward promoter eQTLs, we conducted a stratified analysis of our caQTL-eQTL coloc by genomic context (promoters, intergenic regions, gene bodies, etc.). When examining colocalization rates with DICE eQTLs across these categories, we observed consistently similar levels of overlap regardless of genomic location.

This suggests that our observations are not driven by underpowered eQTL detection at enhancer elements. These analyses are now described in **Supplementary Notes**.

5) Line 375: If the colocalization results between the caQTL and COVID eQTL were largely underwhelming, is it worth reporting/including?

We appreciate this important point. Both the DICE and COVID-19 eQTL resources offer unique advantages and limitations. The DICE dataset, derived from bulk RNA-seq of healthy donors, provides greater statistical power to detect weaker eQTL effects, resulting in stronger colocalization with our sc-caQTLs. However, its biological context differs from our study, as it lacks disease-state samples and may have cell type annotation discrepancies with our scATAC data.

In contrast, the COVID-19 eQTL data shares greater biological relevance with our study, as it comes from a superset of our scATAC samples. While we performed COLOC analysis with both datasets, we observed that the COVID-19 eQTLs showed reduced power compared to DICE, yielding fewer colocalizing caQTLs. This led us to primarily use DICE for downstream analyses while still reporting COVID-19 colocalization results.

Ultimately, we believe the COVID-19 eQTL data provides valuable complementary insights, particularly for COVID-19-specific findings. For instance, we identified several compelling cases where COVID-19 eQTLs colocalized with COVID-19 GWAS signals, offering biologically relevant mechanistic hypotheses. More specifically, we looked at genes that colocalized with our caQTLs in COVID-19 eQTLs but not DICE eQTLs in a cell-type agnostic fashion. In total, we found 194 such COVID-19 eGenes, including *SCAMP1* that is described to be specific in COVID-19 monocytes, but not healthy monocytes in our companion manuscript (Randolph et al.). The possible reasons for the *SCAMP1* eQTL-caQTL colocalization include (1) a COVID-19-specific caQTL that is present in our results, or equally likely, (2) that the caQTL is shared between COVID-19 and healthy monocytes, but additional layers of gene regulation make the eQTL disease-specific. We believe future studies may address these questions more thoroughly. These have been added in our **Supplementary Notes**.

6) Line 393: MS4A4A monocyte eQTL: did you do any analysis to colocalize among DICE eQTL to determine eQTL sharing?

Initially we did not perform colocalization among DICE eQTLs across cell types. In the case of *MS4A4A* eQTL, they don't colocalize because the eQTL is only significant in classical monocytes, and not in CD8 T cells. To fully address this, we performed colocalization between classical monocytes and CD8_STIM for all eGenes included in **Figure 4d**. We found that when a shared caQTL between CD8 T cells and monocytes also colocalized with the same eGene in both cell types, the eQTLs also tend to colocalize. However, this is less likely to be true when the shared caQTLs only colocalize with eQTL in one cell type.

We updated **Figure 4d** accordingly.

7) Line 429: why not look for ABC linked genes?

We appreciate this valuable suggestion. To address it, we conducted a new analysis using the ABC model to identify potential target genes of non-colocalizing cPeaks in CD8+ T cells and monocytes, comparing these results with our original findings. While we identified 188 and 380 putative target genes for non-colocalizing cPeaks in CD8+ T cells and monocytes respectively, these gene sets showed no significant GO biological pathway enrichment (q -value < 0.05). This suggests the ABC model may not effectively capture biologically relevant gene targets in this specific context.

Our original approach focused on TF motif enrichment for these non-colocalizing cPeaks based on the following biological rationale: these regulatory elements may be in a "primed" state, ready to influence gene expression under different cellular conditions. Transcription factor binding patterns can reveal this priming potential, making motif analysis particularly informative for understanding the latent regulatory capacity of these elements. The ABC model, while powerful for identifying active regulatory connections, may be less suited to detect these context-dependent, primed regulatory relationships that could become functional in alternative biological states. We have decided to not add these new analyses to the main text.

8) Line 460: include coloc PPs for caQTL/eQTL

9) Line 467: include coloc PPs for caQTL

We updated the text accordingly (lines 510):

Among GWAS loci that colocalizes with both caQTLs and eQTLs, we found a rheumatoid arthritis-associated variant⁴⁶ near *PVRIG* (also known as *CD112R*), which colocalized with a caQTL in CD4 T cells (chr7:99818835-99819335, PP4=0.96) and eQTLs in T follicular helper cells (PP4=0.94) and naïve Tregs (PP4=0.93). The convergence of chromatin accessibility and gene expression signals at this locus suggests that genetic regulation of *PVRIG*, a T cell co-inhibitory receptor, contributes to rheumatoid arthritis risk, likely through modulation of T cell activation (**Figure S5d**).

In contrast, some GWAS loci only colocalized with eQTLs, revealing potentially distinct regulatory mechanisms. An example was the *RPS26* locus associated with rheumatoid arthritis (chr12:56470625), which showed strong eQTL colocalization across all immune cell types (PP4=0.93–0.95) but no significant caQTL colocalization (PP4=0.19 in B cells). While the *RPS26* promoter was accessible in all cell types, the lack of caQTL colocalization suggests that alternative regulatory processes, such as post-transcriptional mechanisms⁴⁸, may mediate its association with disease (**Figure S5e**).

10) Line 478: "we focused on cPeaks overlapping with TSS": Why was this criteria imposed?

Our approach is motivated by the fact that when GWAS loci colocalize with caQTLs but not eQTLs, identifying their functional target genes becomes particularly challenging. By focusing specifically on cPeaks located at transcription start sites (TSS), we have greater confidence in linking these regulatory variants to their most likely downstream genes. Importantly, this strategy remains effective even when the colocalizing GWAS SNPs are located far from the TSS, as chromatin accessibility at promoters often reflects regulatory effects of a SNP regardless of its precise genomic position. We have clarified this in the revised manuscript on line 538:

In total, 220 (15.3%) caQTL-only loci mapped to promoter regions, which allowed us to nominate a putative causal gene.

11) Line 481: For consistency, include coloc PPs

We updated the text accordingly (lines 539):

In CD4 T cells, the same caQTL colocalized with multiple GWAS including CD, ulcerative colitis (UC), RA and MS (PP4: 0.61-0.84), suggesting it may affect a common mechanism underlying multiple autoimmune diseases. *ZFP36L1* is an RNA-binding protein that is differentially expressed in osteoarthritis and coeliac disease and has been shown to regulate T cell and B cell development⁵⁰⁻⁵², although its exact mechanism in other autoimmune diseases remains unknown (Figure S5f). In another case, the promoter peak (chr22:37256806-37257306) for gene *NCF4* has a caQTL colocalizing with an RA GWAS locus in NK and CD8 T cells (PP4: 0.78-0.80, Figure S5g).

12) Line 495: How was dynamic caQTL colocalization performed?

Thank you for pointing out the absence of this methodological detail. We did not perform colocalization using the results from the interaction model, as that analysis was limited to testing interactions at lead SNPs. Instead, we assessed colocalization using our linear sc-PME model. This strategy is commonly used in similar studies (Nathan et al., Nature 2022; Valencia et al., bioRxiv 2024). We revised the text and Methods section to clarify this point. We also note that statistical methods enabling direct colocalization of dynamic QTLs along trajectories are still under development (e.g., Qi et al., bioRxiv 2025), and we look forward to future improvements in this area.

We updated the Methods accordingly (lines 998):

Since we only have the interaction effects for lead caQTL SNPs, we annotated the colocalization states of dynamic caQTLs by examining the results from the linear sc-PME model.

13) Line 524: Conclusion not well supported. caQTL (COVID patients) and eQTL (DICE - presumably healthy) were not identified in the same context, so other possible explanation besides pleiotropy.

We respectfully disagree on this point. The interpretation of GWAS pleiotropy can be assessed independently for caQTLs and eQTLs, so the fact that they were mapped in different samples does not undermine our conclusions.

For caQTLs, we observed that GWAS loci colocalizing in more contexts tend to associate with a greater number of distinct cPeaks across those contexts (**Figure S6a**). This suggests that pleiotropy complicates the identification of the causal cPeak and cell type for a given GWAS locus. A similar trend was seen for eQTLs (**Figure S6b**), indicating that determining the true causal eGene and cell type is equally challenging. Initially, our analysis focused only on GWAS loci colocalizing with either eQTLs and caQTLs, but we now show that this pattern holds for GWAS loci that colocalized with both eQTLs and caQTLs.

Thus, our interpretation of GWAS pleiotropy remains valid regardless of differences in sample sources between caQTL and DICE data. In fact, the same conclusion could be drawn from analyzing either caQTLs or eQTLs alone, but we believe that the inclusion of both strengthens our findings. We have clarified this in the revised manuscript on line 581:

Furthermore, GWAS loci with multiple context colocalizations typically affected distinct chromatin peaks (cPeaks) and eGenes across these contexts (**Figure S6a,b**). This pattern persisted regardless of whether loci colocalized with caQTLs alone, eQTLs alone, or both, indicating that pleiotropy similarly complicates interpretation of both caQTL and eQTL colocalization analyses. These findings highlight a fundamental challenge in GWAS functional interpretation: the widespread pleiotropy of regulatory variants, which frequently influence distinct genes and chromatin regions across different cellular contexts. Thus, additional layers of evidence - beyond simple colocalization - may be necessary to confidently assign causal mechanisms to GWAS loci.

14) Figure 1g: would be good to include study information in addition to disease state so that can see if study was related to groupings

We updated the figure accordingly.

15) Figure 2b: what is the dashed box?

The dashed box represents topics that do not clearly represent cell types. We updated the legend to make this clearer:

b. Heatmap showing the average loading for each topic in each cell-type in L2 annotation. Dashed box indicates topics without clear enrichment in any cell types.

16) Figure 2d: Typo? Is it supposed to be k1 instead of k2?

Thank you for noticing this. We updated the legend to fix this error.

17) Figure 4a: the q-value thresholds used to determine cell type specific caQTL are confusing. $q < 0.1$ is considered a significant caQTL in a given cell type, but if the q-value is > 0.025 in other cell types, the caQTL is considered cell type specific even though it would technically considered significant in the other cell type.

Thank you for raising this point. You are correct that caQTLs with $0.025 < qval < 0.1$ in another cell type are technically significant. Our choice of thresholds stems from multiple hypothesis testing adjustments. While $qval < 0.1$ defines significant caQTLs in a given cell type, assessing cell-type specificity requires further correction for the number of cell types analyzed.

Since we focused on five major cell types (NK and other T cells had very few caQTLs), we applied a Bonferroni correction: 0.1 divided by (5-1) = 0.025. This means that for a caQTL with $qval < 0.1$ in cell type A to be considered shared, it must have $qval < 0.025$ in another cell type. This stringent approach yields a more conservative estimate of shared caQTLs compared to using $qval < 0.1$ alone.

Despite this conservative threshold, we still observed widespread sharing of caQTLs across immune cell types (except monocytes), supporting our conclusion that caQTL sharing is pervasive. We have added this rationale in the revised **Supplementary Notes**.

18) Figure 4b: top plot, why use log₁₀ scale?

We chose log₁₀ scale as the number of eQTLs across groups can differ by two orders of magnitude (e.g. the left most versus the right most bar). Using a linear y-axis would make the shortest bar invisible. However, even in the shortest bar (caQTL shared in 7 cell types), we wanted to highlight that sc-PME model found more caQTL than RASQUAL.

19) Figure 4 legend: d and e flipped

We updated the legend to fix this error:

d, Top: heatmap for COLOC PP4 of shared caQTLs between CD8 T cells and monocytes. Bottom: example Manhattan plots for shared caQTL colocalizing with cell-type specific eQTLs in monocytes and CD8 T cells, respectively. **e**, Logistic regression coefficients and 95% confidence intervals for variables that predict caQTL-eQTL colocalization. Model includes all caQTL-eQTL pairs that colocalize in at least one cell-type.

20) Figure 5g: Where is the NK caQTL locus plot? Mentioned to colocalize in main text (line 549).

We initially did not include them because the central message in **Figure 5g** is for monocytes. We have now added the LocusZoom plots for the caQTLs in CD4 T cells (not NK cells) mentioned to the **Supplementary Figure**.

21) Figure 5 legend: missing description for plot e.

We updated the legend to fix this error:

e, Schematic representations for testing the utility of eQTL and caQTL in mapping putatively causal genes for GWAS. GWAS loci were grouped by their colocalization with either eQTL, caQTL or both, in the same cell type context or not. GWAS loci that only have caQTL are mapped to the nearest gene.

22) Extended Figure 1d: include study information in addition to disease state?

We updated the figure accordingly.

23) Extended Figure 5c: Are these eQTL or caQTL signals?

The signals in **Figure S5c** are for caQTLs in *RPS26* TSS peak. We updated the figure labels and legends accordingly.

24) Extended Figure 5e: Include plot for CD8? Mentioned in text (line 487)

We updated the figure accordingly.

25) Extended Figure 6: An opportunity for an UpSet plot to better visualize sharing?

In **Figure S6**, we wanted to highlight the trend that the number of contexts in which a GWAS locus is colocalized with eQTL/caQTL (x-axis) is positively correlated with the number of colocalized eGenes/cPeaks across all contexts (y-axis), suggesting that pleiotropic effects hinders biological interpretation. **We modified both the main text and figure legend to better convey this message.**

26) Some general comments about the all of the LocusZoom plots in main and extended figures:

- x-axes should be aligned for all stacked plots for easier interpretation
- Similarly, in "shared" cases (i.e. shared caQTLs/eQTLs), y-axes should be standardized
- For stacked plots, a single index variant should be selected and highlighted in all plots so that LD variants are colored consistently for easier interpretation.
- Include an LD color legend

We updated all LocusZoom figures according to the above comment a) and d). We found that using a single index variant made the plots more difficult to interpret and so did not opt to address c). Instead, we keep the vertical dashed lines to indicate the index variant from each data type in all plots.

e. Gene tracks should be added to include the target gene or eGene and also give a reference for where cPeaks are relative to the gene.

We have this information for most of the LocusZoom plots in the manuscript. We added this information where it is currently missing (e.g. Fig 3h, Fig 4d).

f. When peak tracks are included, add y-axis

We updated all peak tracks accordingly.

g. For all LocusZoom plots include cell type and analysis (i.e. caQTL/eQTL)

We updated all LocusZoom figures to make cell type and analysis more obvious.

Reviewer #2: This manuscript provides a comprehensive study on chromatin accessibility QTLs (caQTLs) across immune cell types, utilizing single-cell ATAC-seq data. The authors constructed a harmonized chromatin accessibility map from 282,424 cells across multiple conditions, identified extensive caQTLs and analysed their relations with eQTLs and GWAS loci. The study sheds light on the functional effects of caQTLs and their relevance to interpret disease associated genetic variants, emphasizing the need to expand eQTL studies in disease-relevant contexts. The computational approaches are rigorously designed and the results look solid, leading to some novel insights into the quantitative understandings of the genetic basis of gene regulation under specific cellular contexts. I just have a few questions for the authors to further clarify.

We thank you for the positive comments and questions.

Major Comment:

1. The inferred topics from topic models are not necessarily cell types or subtypes. The topics may correspond to variations of co-factors and batch effects. The authors need to provide a justification to clear this potential issue.

Thanks for this comment. We agree that some topics can reflect technical factors and batch effects. To better understand this, we first calculated correlations between topic loadings and continuous measurements of batch and data quality, including MTratio, ReadsInTSS and TSSEnrichment etc. Several topics showed correlations with these QC metrics. Notably, most of these topics do not show cell-type-specific enrichment (k2, k5, k9, k16, k19; **Figure 2b**). We also calculated mean topic loadings in each dataset and found that most topics are not enriched in any specific dataset. This shows that batch effects are captured by certain topics and have minimal impact on other biologically-relevant topics. We believe this is a desirable property for topic modeling because topics that reflect batch effects can be excluded from downstream analysis. Furthermore, we also regressed out batches and QC metrics in all our downstream analysis, similar to previous methods (Mitchel et al., Nature Biotech., 2024). Therefore, we believe that batch effects do not confound our topic analysis.

We added the following figures to **Figure S3**:

2. About the colocalization analysis of caQTLs and eQTLs, could the authors show the distribution of their effect sizes respectively and check whether there is a bias caused by smaller effect sizes of subsets of non-overlapping QTLs?

As expected, we found that colocalized caQTLs tend to have the largest Z scores, followed by uncolocalized caQTLs. This difference is larger in B cells, which has the smallest number of cells, therefore lower statistical power. This observation is consistent with our previous eQTL-GWAS analysis (Mu et al., *Genome Biology*, 2021). However, many uncolocalized caQTLs showed Z scores that are comparable to colocalized caQTLs. This means that power only partially explains the lack of colocalization between caQTL and eQTL.

We added this analysis to Supplementary Notes.

3. For caQTLs that do not colocalize with eQTLs, how is the genomic distance distribution between these caQTLs to their nearest gene promoters?

We found that, as expected, caQTLs colocalized with eQTLs are closer to TSS compared to those that do not colocalize. However, this difference is small, and we did not observe enrichment of colocalized cPeaks in gene promoters compared to uncolocalized cPeaks. We have added these additional analyses in the Supplementary Notes.

4. For cell-type specific caTLA[caQTL]-eQTL colocalizations (versus shared caQTL-eQTL), in addition to the enrichment with lineage-associated TF motifs, I wonder whether the authors could check whether specific TF motif combinations are enriched? The combinatorial TF patterns may represent the regulatory grammar for precise cell-type specific regulation.

Thank you for your thoughtful suggestion. Studying TF co-binding could indeed provide even more precise insights into cell-type-specific regulatory grammar. However, given the limited number of cell-type-specific caQTL-eQTL colocalizations in our current dataset, we worry that such an analysis might not yield statistically meaningful combinations at this stage.

That said, we hope you'll agree that the cell-type-specific enrichment of TF motifs in colocalized cPeaks still offers compelling and informative evidence for a plausible regulatory mechanism.

5. On Page 10, lines 399-417: it is confusing to interpret the importance of caQTL-eQTL pair colocalize in multiple cell types as the strongest feature to predict cell-type specific caQTL-eQTL colocalizations. If it is cell-type specific, why does the multi-cell-type sharing feature become the strongest feature?

Thank you for pointing out this crucial detail. In this analysis, we found that colocalization in multiple cell types is a strong predictor between CD4 and CD8 T cells. This is due to the higher level of caQTL-eQTL sharing among these cell subtypes. We have added a **Supplementary Figure** as shown below. This predictor has smaller effects in monocytes (Figure 4e), for instance, since most caQTL-eQTL colocalization in monocytes are cell-type-specific. This is largely due to the lack of eQTLs, but not caQTLs across cell types (Figure 4d). It is also because in the logistic regression

we used all caQTL-eQTL pairs in COLOC, and many of them did not colocalize in any cell type. Including them in the model also makes effect sizes for multi-cell-type COLOC larger.

Minor Comment:

1. Some annotations in the manuscript remain unclear, such as the terms "(MONO%, MONO#, GRAN%MYELOID)" on page 5, which need further clarification or definition. Additionally, there are still some typographical and grammatical errors throughout the text that should be addressed to improve readability and professionalism.

Thank you for pointing out these details. We added the full names of these GWAS phenotypes (MONO%: Monocyte percentage of white cells, MONO#: Monocyte count, GRAN%MYELOID: Granulocyte percentage of myeloid white cells) to the main text. We have also provided a **Table S13** listing the full names of all GWAS phenotypes shown in **Figure 2d**.

We corrected all the typos and grammatical errors throughout the text.

2. The choice of colors in some figures, such as Figure 3d and Figure 3j, makes it difficult to discern finer details. Enhancing the contrast and clarity of these figures would significantly improve their interpretability and overall quality.

We updated these figures to make sure they are clear and legible.

Reviewer #3: Overview: Mu et al identify and analyze chromatin accessibility (ca)QTLs using harmonized datasets of single-cell ATAC data from 59 samples (48 unique donors). This work has two main contributions. Firstly, they provide a pipeline to harmonize cell type annotations across multiple data sources, followed by topic analysis to demonstrate patterns of chromatin accessibility across cell types and cell states. Second, the authors perform caQTL mapping and systematically evaluate colocalization with eQTLs and GWAS loci for relevant traits. Their approach and results are highly likely to be of interest to the statistical genetics/genomics communities and provide insights on interpreting GWAS risk loci using multi-omics approach. This work has important implication on the lack of cellular context in eQTL mapping and more robust causal inference for GWAS risk loci when observing convergence of both modalities. The manuscript is written well and results presented clearly. With that said I have several comments.

Thank you for the positive comments.

Major Comments:

1. Regarding the COVID-19 example, the author identified two groups of COVID-19 patients have more NK or monocyte cells as presented in Figure 1g. However later when using topic analysis, the authors concluded the k17 topic represent CD8 TEM and its loading tracks with COVID-19 cells proportions, thus suggesting an expanded CD8 TEM population in COVID-19 patients. The authors should comment on why NK or monocyte expansion is absent in the topic analysis and why this discrepancy can occur.

Thank you for raising this insightful question. We'd like to clarify that while certain cell populations may expand during COVID-19 infection, this doesn't automatically translate to higher topic loadings in those populations. A cell type could increase in number while maintaining similar topic usage patterns as non-expanded populations.

To explore this relationship systematically, we first verified through cluster analysis that CD8 TEM cells are indeed expanded in COVID-19 patients. We then specifically examined the association between COVID-19 status and topic loadings in two key populations: monocytes (k15) and NK cells (k8). Our analysis revealed that while monocyte topic 15 loadings showed a strong positive association with COVID-19 status ($\beta=1.13$, $p=7.99 \times 10^{-16}$), NK cell topic 8 loadings demonstrated no significant association ($\beta=-0.15$, $p=0.25$).

These findings highlight that topic loadings and cell population proportions provide complementary but distinct biological information. Our focus on CD8 TEM topic 17 was motivated by its particularly strong enrichment for COVID-19 heritability, which we believe offers unique insights into disease mechanisms. We added these results to **Figure S3**.

2. I suggest the authors first describe ancestry, age, and sex for in-sample individuals to provide the generalizability context of results produced by this work, especially for colocalization results with external dataset (e.g., COVID19-patients in-sample vs. all healthy individuals in DICE dataset). For example, the PCA plot shows the 20 individuals collected by this study are either European- or African-ancestry. Based on accompanied paper Randolph et al, these 20 individuals are all males.

We updated the text accordingly and added the following few sentences (line 97):

To build a detailed chromatin accessibility atlas of peripheral blood mononuclear cells (PBMCs), we integrated single-cell ATAC-seq (scATAC-seq) data from multiple sources, including 20 hospitalized COVID-19 patients (with 8 sampled during convalescence), 5 healthy controls from our study, and 23 additional samples from published datasets. These comprised 13 healthy controls from Benaglio et al.⁹ and 8 COVID-19 convalescent donors plus 2 healthy controls from You et al.¹⁰, all meeting consistent quality metrics (TSS enrichment and fragments per cell; **Figure S1a,b**). The combined cohort (Table S1) represents diverse ancestries, including European, African, admixed American, and East Asian backgrounds—the latter exclusively from the You et al. dataset (Figure 1a). Our study and You et al. collected data from both COVID-19 patients and healthy controls, enabling comparative analyses across disease states and recovery.

3. For the caQTL mapping, the authors found extensive sharing of caQTLs using Poisson mixed effect in contrast to more cell-type specific results using RASQUAL. Since the sample size is relatively small and peak counts are sparse in this analysis, the authors should demonstrate that PME model is well-calibrated under null by permuting peak counts and show distribution of p values. The authors should also explain why age, sex and COVID-19 disease status are not considered in the caQTL model.

Thank you for your comment. To address the computational challenges while demonstrating that our PME model is well-calibrated, we performed permutation analyses focusing specifically on chromosome 1 in monocytes from COVID-19 data.

For this validation, we analyzed both RASQUAL-defined cPeaks (which by definition contain significant caQTLs) and carefully matched non-cPeak controls with similar sparsity and mean counts. Our permutation tests showed excellent calibration: For cPeaks, 9.55% had q-values < 0.1 under permutation. For matched non-cPeaks, 8.50% had q-values < 0.1 under permutation. These results closely match the expected null distribution for both tested SNPs and lead SNPs, confirming our model's proper calibration.

Regarding potential confounding by sex and age, we agree these are important considerations. Following standard practices in the field, we account for these factors by including phenotypic PCs in our analyses. This approach has been widely adopted in QTL mapping studies.

We added the following sentences in the main text (line 344):

Furthermore, sc-PME results strongly agree with those obtained using pseudo-bulk data and QTLtools. The method proved robust in permutation tests and achieved greater statistical power by explicitly modeling both intra- and inter-individual variation at single-cell resolution. These findings, detailed in the **Supplementary Notes**, collectively establish sc-PME as a reliable and powerful approach for single-cell QTL mapping that maintains consistency with established methods while providing higher sensitivity.

4. The authors used logistic model to predict colocalization between caQTL and eQTLs using multiple predictors, but not including TSS distance. Can the authors plot and describe the distance of lead caQTL to TSS of colocalized gene (i.e., eQTLs) and for those not colocalized respectively? Assuming promoter effect of eQTL is likely shared between cell types, I would expect the majority of colocalized ones are close to TSS. The non-colocalized ones are likely distal where eQTL signals are lacking due to limited eQTL sample size or lack of relevant context.

We performed the analysis as you suggested. As expected, colocalized cPeak tend to be closer to the nearest TSS. However, many non-COLOC caQTLs have comparable distance to TSS as COLOC caQTLs. Therefore, we believe that the difference in genomic locations of eQTL and caQTL cannot fully explain the lack of colocalization between the two. This result was added to **Supplementary Notes**.

5. The in-sample COVID-19 eQTL mapping from Randolph et al. was performed for controlled samples and IAV-infected samples respectively. Can the authors clarified on which results are used in this work? If using both, the authors should consider present their colocalization results between caQTLs and these condition-specific eQTLs separately.

Because of the smaller number of eQTL from Randolph et al., we performed colocalization analysis in control and IAV-infected samples separately, and then reported the union set of colocalization genes. We separated them in **Figure S5**. In most cell types, Baseline condition has the most colocalization events, because it has the largest sample size. The only exception is monocyte, which has been shown to respond to COVID-19 infection in Randolph et al.

COLOC of caQTL with COVID-19 eQTL

Minor Comments:

1. For Figure 1g, it's not clear which two groups of individuals have expanded NK or monocyte population. Can the authors highlight those groups as done in Extended Figure 1d?

We updated the figure accordingly with dashed boxes.

2. For Figure 3i, consider order the cell type on x-axis by their total number of cells to show their relationship with number of identified dynamic caQTLs.

We updated the figure accordingly.

3. For the top bar plot in Figure 4c, the y axis name should be "Number of colocalized caQTLs".

We updated the figure accordingly.

4. The authors should clarify on the interpretation on Figure 4d and 4e as it shows mixed message as written in line 407-409. The cell-type specific colocalization in Figure 4d is likely driven by difference in lymphocytes (eg. T cells) versus monocytes. The strong predictor of "COLOC in another cell type" in Figure 4e is likely reflecting similar subtypes as CD4 T cells and CD8 T cells.

Thank you for pointing out this crucial detail. We agree that the predictor "COLOC in another cell type" could reflect T cell subtypes. Indeed, the effect size for this predictor is the largest between CD4 and CD8 T in **Figure 4e**, reflecting higher levels of similarity between CD4 T and CD8 T. However, this predictor is also significant in other cell types (such as monocytes), albeit having smaller effects, suggesting that at least some colocalization signals are shared between monocytes and non-monocytes, including CD8 T, as in **Figure 4d**. To reconcile these seemingly contradictory interpretations, we added a Supplementary Figure showing the COLOC statistics for all caQTL-eQTL pairs that are colocalized across all cell types. Indeed, we found that the

majority of shared caQTL-eQTL pairs are within CD4 T subtypes. We hope this will provide a more complete picture of the sharing of colocalization among cell types. We also revised the main text to make this less confusing.

5. For caQTL mapping in the 25 individuals with repeated measures collected by this study, can the authors clarify on how different time point are modeled? Or did the authors use one time point?

Thank you for pointing this out. To avoid repeated measurements, we only used the time point that has the most cells to generate pseudobulk counts in RASQUAL caQTL analysis. For the single-cell PME model, we used random effects to account for repeated measurements and therefore used all cells. We updated the Methods section to state this explicitly (line 937:).

Because a subset of our COVID-19 samples are collected from the same individuals at two visits, we only included the time point with more cells to avoid repeated measurement in RASQUAL model.

6. Figure 5e lacks description in the caption.

We updated the legend accordingly:

e. Schematic representations for testing the utility of eQTL and caQTL in mapping putatively causal genes for GWAS. GWAS loci were grouped by their colocalization with either eQTL, caQTL or both, in the same cell type context or not. GWAS loci only have caQTL are mapped to the nearest gene.

Referees' report, second round of review

Reviewer 1:

The authors have fully addressed my previous concerns, and I commend them for a comprehensive revision. I have no remaining concerns.

Reviewer 2:

The authors have addressed all my original comments. I recommend acceptance of this study for Cell Genomics.

Reviewer 3:

I thank the authors for the considerable effort in addressing my comments. I have no new comments at this time.

Authors' response to the second round of review